# Evaluation of the Robustness Verification of Downstream Production Process for Inactivated SARS-CoV-2 Vaccine and Different Chromatography Medium Purification Effects

**DOI:** 10.3390/vaccines12010056

**Published:** 2024-01-06

**Authors:** Jia-Hui Pang, Chang-Fu Guo, Peng-Liang Hao, Sheng-Li Meng, Jing Guo, Dou Zhang, Ya-Qi Ji, Ping-Gang Ming

**Affiliations:** 1National Engineering Technology Research Center for Combined Vaccines, Wuhan 430207, China; 2Wuhan Institute of Biological Products Co., Ltd., Wuhan 430207, China

**Keywords:** SARS-CoV-2, Vero cells, downstream process, purification, vaccines, size-exclusion chromatography, anion-exchange chromatography

## Abstract

Background: Large-scale vaccine production requires downstream processing that focuses on robustness, efficiency, and cost-effectiveness. Methods: To assess the robustness of the current vaccine production process, three batches of COVID-19 Omicron BA.1 strain hydrolytic concentrated solutions were selected. Four gel filtration chromatography media (Chromstar 6FF, Singarose FF, Bestarose 6B, and Focurose 6FF) and four ion exchange chromatography media (Maxtar Q, Q Singarose, Diamond Q, and Q Focurose) were used to evaluate their impact on vaccine purification. The quality of the vaccine was assessed by analyzing total protein content, antigen content, residual Vero cell DNA, residual Vero cell protein, and residual bovine serum albumin (BSA). Antigen recovery rate and specific activity were also calculated. Statistical analysis was conducted to evaluate process robustness and the purification effects of the chromatography media. Results: The statistical analysis revealed no significant differences in antigen recovery (*p* = 0.10), Vero HCP residue (*p* = 0.59), Vero DNA residue (*p* = 0.28), and BSA residue (*p* = 0.97) among the three batches of hydrolytic concentrated solutions processed according to the current method. However, a significant difference (*p* < 0.001) was observed in antigen content. Conclusions: The study demonstrated the remarkable robustness of the current downstream process for producing WIBP-CorV vaccines. This process can adapt to different batches of hydrolytic concentrated solutions and various chromatography media. The research is crucial for the production of inactivated SARS-CoV-2 vaccines and provides a potential template for purifying other viruses.

## 1. Introduction

Novel Coronavirus Pneumonia (COVID-19) is a global infectious disease caused by the novel coronavirus (SARS-CoV-2) [1]. This pandemic has left an indelible mark on the world, with millions of lives lost and significant economic repercussions [2]. Effective vaccines have emerged as a pivotal tool in our battle against this virus [3]. Vaccination not only provides substantial health benefits, reducing mortality rates and enhancing public health [4], but it also carries profound societal and economic advantages. Vaccination can mitigate healthcare costs, stimulate economic growth, and boost overall productivity [5]. Recent research underscores the risks associated with reinfection, as a second bout of COVID-19 can result in symptoms of similar severity to the initial infection [6]. Furthermore, multiple or prolonged infections may lead to cognitive impairment [7]. Regular vaccination stands as a powerful defense against the risk of reinfection, underscoring the continued need for research and vaccination efforts against the COVID-19 virus [8].

In vaccine production, downstream purification processes are pivotal [9]. These processes involve a series of steps designed to separate and refine intermediate products generated in the earlier stages [10]. By eliminating various impurities from these intermediates, the final vaccine product suitable for administration is achieved. Chromatography stands as a well-established and widely used method in virus purification [11]. Size exclusion chromatography (SEC), a technique based on the size of molecules, is particularly effective. It enables the separation of viruses of different sizes by selecting filters with varying pore sizes [12]. SEC has demonstrated its efficacy in purifying a range of viruses, including turkey coronavirus [13], vesicular stomatitis virus [14], Moloney murine leukemia virus [15], and equine influenza virus [16]. While SEC excels in removing impurities such as proteins, it can be challenged by the need for efficient removal of host cell DNA, thus prompting the use of anion exchange chromatography (AEC) for effective hcDNA clearance [17,18]. Challenges persist, including the virus’s sensitivity to process conditions, such as shear stress, temperature, pH, and UV light [19]. Additionally, multiple factors must be considered when designing downstream processes, encompassing buffer systems and the sequencing of operations [20]. Even minor alterations can have profound implications for subsequent outcomes. Despite these complexities, our extensive experience in vaccine production and research enabled us to swiftly develop a combined SEC and AEC downstream purification process for SARS-CoV-2 during this pandemic. This approach proved successful, with our vaccine candidate, WIBP-CorV, demonstrating robust immunogenicity and safety in clinical trials [21]. The reason for choosing the inactivated vaccine technology was that its development took place in the early stages of the pandemic when mRNA vaccine technology had not yet been widely promoted. Therefore, we opted for this technology at that time. Additionally, we closely monitored various new vaccine technologies. Subsequently, we also pursued the development of mRNA vaccines for both influenza and the novel coronavirus.

To gauge the robustness and repeatability of our existing downstream purification process, we conducted an experiment utilizing three distinct batches of the Omicron strain COVID-19 hydrolytic concentrated solution. It is noteworthy that inherent variations in the properties and quantities of the hydrolytic concentrated solution among batches can potentially disrupt downstream purification processes. To achieve a comprehensive evaluation of robustness, we introduced a significant element of diversity. Chromatography media from four different companies were separately integrated into the purification processes. The effectiveness of the purification process was assessed by measuring several key quality parameters in both the initial and final purification samples. These parameters included Vero cell DNA and protein levels, antigen content, total protein content, residual bovine serum albumin (BSA), and antigen recovery rate. A thorough statistical analysis of the collected data was subsequently conducted to ensure a comprehensive assessment of the current downstream process’s robustness.

## 2. Materials and Methods

### 2.1. Cell Culture Conditions and Virus Production

The Vero cells used in this experiment were obtained from the newly emerging infectious diseases laboratory and were cryopreserved. The viral strain used was the Omicron B.1.1.529 strain of SARS-CoV-2, initially isolated from clinical samples (bronchoalveolar lavage fluid) from Hong Kong, and inoculated into Vero-E6 cells in the P3 laboratory. The process began with the revival of Vero cells stored in liquid nitrogen, which were subsequently subjected to multiple passages in Corning^®^ 225 cm^2^ cell culture flasks (CORNING, New York, NY, USA) to obtain a sufficient quantity of live cells. The microcarrier and Vero cells were co-cultured in a 150 L bioreactor (Sartorius AG, Otto-Brenner-Str. 20, 37079 Göttingen, Germany). Following an initial period of cultivation, the culture was further scaled up to a 1000 L bioreactor (Sartorius AG, Otto-Brenner-Str. 20,37079 Göttingen, Germany). Specific operating parameters for the bioreactor can be found in the manufacturer’s instructions. The bioreactor was operated with the following settings: temperature: 37 °C, dissolved oxygen (DO): 70%, pH: 7.20, rotary filter speed: 50 rpm, agitation speed: 30 rpm, vessel pressure: 0.05 bar, surface air flow rate: 20.0 L/min. Following cell cultivation, the working seed batch virus dose was calculated based on cell count results. The virus titer was 6.0 log CCID50/mL, and the multiplicity of infection (MOI) for inoculation was set at 0.0001. Table 1 displays the upstream operational conditions and quantification.

### 2.2. Harvest and Inactivation

The virus was harvested upon reaching the desired cytopathic effect (CPE), with the maximum pressure set during harvesting not exceeding 1.0 bar. In compliance with biosafety measures, immediate inactivation procedures were employed post-harvest. β-propiolactone was utilized as the inactivating agent, and the quantity of this agent was calculated based on the volume of the harvested fluid, followed by a thorough mixing process. After the inactivation is completed, a portion of the samples should be extracted for inactivation verification. This involves transferring the inactivated samples into Vero cells and culturing them for three consecutive generations. The purpose is to observe any signs of pathology and conduct sequencing to ensure that complete inactivation has been successfully achieved before proceeding with subsequent steps.

### 2.3. Clarification and Concentration

Both microfiltration and ultrafiltration procedures were performed using Sartorius Cross Flow Ultra-filtration Systems (Sartorius AG,Otto-Brenner-Str. 20,37079 Göttingen, Germany). Before usage, the equipment was meticulously cleaned with a 0.5 M NaOH wash, followed by thorough rinsing with sterile filtered purified water. The maximum pressure setting was maintained at 0.1 Mpa. For microfiltration, Hydrosart^®^ microfiltration membranes with a pore size of 0.22 μm and a membrane area of 200 cm^2^ were employed. In the case of ultrafiltration, Hydrosart^®^ ultrafiltration membranes from Sartorius (Germany) with a molecular weight cutoff (MWCO) of 300 kDa and a filtration area of 0.7 m^2^ were utilized.

### 2.4. SEC

The SEC purification was carried out using the Äkta™ avant chromatography system from GE Healthcare Life Sciences. This system was equipped to monitor UV absorbance at 280 nm and conductivity at room temperature. Four distinct gel filtration media were employed for this purification process: Chromstar 6FF (Bio-Link, Shanghai, China), Singarose FF (HUACHUN BIOLOGICAL, Tianjin, China), Bestarose 6B (BESTCHROM, Shanghai, China), and Focurose 6FF (HUIYAN Bio, Wuhan, China). The buffer solution was pre-filtered using BIOFIL^®^ Syringe Driven Filters (0.22 μm, Guangzhou, China) before use and degassed using a digital ultrasonic cleaner (KQ-3000DV, KUNSHAN ULTRASONIC INSTRUMENTS Co., Ltd., Kunshan, China). In accordance with the manufacturer’s instructions, the gel filtration media were loaded into XK16/100 chromatography columns from GE Healthcare Life Sciences. Once column efficiency exceeded 2500 and symmetry fell within the range of 0.8 to 1.5, the subsequent steps of the experiment were initiated. Initially, the gel filtration media were equilibrated with a pH 7.4 PBS buffer at a flow rate of 1 mL/min. Subsequently, the hydrolytic concentrated solution was introduced into the chromatography column, with each addition being 10% (17 mL) of the column volume. The constituents of all eluates were monitored by measuring the UV absorbance at 280 nm, and the first peak fraction was collected. The specific methods for collecting samples are consistent with the current formal production process. The collection of samples can begin after 0.15 CV, which means no substances should be collected before this point. The collection must start at 0.35 CV, even if no substances are detected at UV 280 nm between 0.15 and 0.35 CV. The collection of samples can be completed after 0.36 CV, which means it must continue until 0.36 CV, even if no substance is observed flowing out during UV 280 nm monitoring before this point. The collection of samples must be completed at 0.56 CV, even if there is still substance flowing out at this point during UV 280 nm monitoring. Following each experiment, the column was regenerated and washed with a 1M NaOH solution until both the conductivity and UV absorbance at 280 nm stabilized, and then washed with PBS solution until both the conductivity and UV absorbance at 280 nm stabilized. Bubbles in the chromatographic column were checked for, and once it was confirmed that there were no issues, the next set of experiments commenced to maintain independence between experimental groups. This procedure was replicated three times for each type of gel filtration medium.

### 2.5. AEC

The AEC experiment was conducted using the Äkta™ avant chromatography system from GE Healthcare Life Sciences. Four distinct anion exchange chromatography media were employed: Maxtar Q (Bio-Link, Shanghai, China), Q Singarose (HUACHUN BIOLOGICAL, Tianjin, China), Diamond Q (BESTCHROM, Shanghai, China) and Q Focurose (HUIYAN Bio, Wuhan, China). In adherence to the manufacturer’s instructions, the AEC media were packed into XK50/30 columns from GE Healthcare Life Sciences. At this point, consistent with the approach taken with the SEC, it is necessary to test column efficiency and symmetry before proceeding with subsequent steps. Perform the next step when the column efficiency is greater than 2500 and the symmetry is between 0.8 and 1.5. Before loading the sample, wash 5 CV with purified water, followed by equilibration with at least 5 CV of PBS buffer until UV 280 nm and conductivity values stabilize. During washing and equilibration, the flow rate can be set to 10 mL/min to 15 mL/min, ensuring that the pressure does not exceed 0.3 MPa. The sample is the liquid collected from SEC, and the sample volume at this time is 30 mL, approximately 6.1% of the CV. In the subsequent process, the flow rate should be set to 5 mL/min. Immediately after loading the sample, switch the liquid to 0.5 M NaCl. The specific method for collecting samples is as follows, in accordance with the current practices in the workshop. The collection of samples can begin at 0.72 CV, which means no substances should be collected before this point, even if there is material detected at UV 280 nm. The collection must start at 0.82 CV, even if no material is detected at UV 280 nm at this time. The collection of samples can be completed at 0.84 CV, which means it must continue until this point, even if no material flows out at UV 280 nm before this. The collection of samples must be completed at 0.97 CV, even if there is still material flowing out at UV 280 nm at this time. Regenerate between each set of experiments using 5 CV of 1 M NaOH solution, followed by washing with 5 CV of purified water, and finally equilibrating with at least 5 CV of PBS buffer until UV 280 nm and conductivity values stabilize to ensure the independence of each set of experiments. Repeat the entire process three times for each chromatographic medium.

### 2.6. Detection of Quality Parameters

#### 2.6.1. Residual DNA Content

The residual DNA content of Vero cells was detected using the PCR-fluorescent probe method with the SHENTEK^®^ Vero Residual DNA Fragment Analysis Kit (HZSKBIO, Huzhou, China). All procedures were conducted as per the manufacturer’s manual.

#### 2.6.2. Vero Cell HCP

The Vero cell HCP was assessed using the Enzyme-Linked Immunosorbent Assay (ELISA) method with the Vero Residual HCP Detection Kit (AnorMed, Beijing, China). Subsequent operations were carried out following the manufacturer’s manual.

#### 2.6.3. BSA Residual Levels

Residual levels of BSA were determined using the ELISA method. The Quantitative Bovine Serum Albumin Enzyme-Linked Immunosorbent Assay Kit (Cat#: RB0001, Bosheng Medical Biotechnology Development Co., Ltd., Wuxi, China) was employed for this purpose. The procedures were performed according to the manufacturer’s manual.

#### 2.6.4. Total Protein Assay

The protein content was quantified using the BCA method. The Micro BCA™ Protein Assay Kit (#23225; Thermo Fisher Scientific (China) Co., Ltd., Shanghai, China) was utilized for protein quantification. All steps were carried out following the manufacturer’s manual.

#### 2.6.5. Antigen Content Detection

The ELISA method was used to measure the antigen content. The standard for the novel coronavirus antigen content was prepared and stored in our institution’s quality assurance laboratory. Antibodies were prepared using serum from rabbits immunized with the novel coronavirus, and HRP Goat Anti-Rabbit IgG (#BA1054, Boster Biological Technology Co. Ltd., Wuhan, China) was used in the experiment. All subsequent operations were conducted according to our institution’s SOP for Novel Coronavirus Inactivated Vaccine (Vero Cell) Antigen Content Detection. The novel coronavirus antigen content standard and samples were diluted to the initial dilution of 200 µL/well, and added to the coated plate in duplicate. Two-fold serial dilutions of not less than 6 dilutions were made in each well. The plate was placed in a constant temperature water bath at 37 °C for 1 h. The plate was washed 5 times with PBST washing solution. The detection antibody was added at the working concentration of 100 µL/well and incubated in a constant temperature water bath at 37 °C for 0.5 h. The plate was washed 5 times again. The enzyme-labeled antibody was added at the working concentration of 100 µL/well and incubated in a constant temperature water bath at 37 °C for 0.5 h. The plate was washed again 5 times. TMB coloration solution was added at 100 µL/well, mixed well, and then placed in a 37 °C water bath for 15 min. After adding the stop solution at 50 µL/well, the OD value was measured at 450/630 nm, and all sample OD values were subtracted from the average OD value of the negative control. The antigen content of the test sample was calculated using the double parallel line method.

### 2.7. Statistical Methods

Differences in quality parameters between different groups were compared using Welch’s *t*-test. A *p*-value less than 0.05 was considered statistically significant. Pearson correlation analysis was employed to investigate the relationships between various quality parameters and process parameters. Statistical analysis and data presentation were conducted using the computer software R version 4.3.1.

## 3. Results

### 3.1. Virus Purification by SEC

Purification was conducted on three separate batches of the Omicron strain COVID-19 hydrolytic concentrated solution utilizing Chromstar 6FF, Singarose 6FF, Bestarose 6B, and Focurose 6FF columns. The characteristics of the packing materials, including bed height, volume, height equivalent to a theoretical plate (HETP), and asymmetry are summarized in Table 2. Each batch of the hydrolytic concentrated solution underwent three repeated experiments in chromatography columns filled with the four types of packing materials. In each experiment, a sample volume of 0.10 CV of the hydrolytic concentrated solution was employed, resulting in a total of 36 SEC purifications. Real-time monitoring of UV absorbance at 280 nm was conducted throughout all experiments. Figure 1 illustrates the UV 280 nm chromatograms of the three batches of hydrolytic concentrated solution after purification using the four SEC packing materials. Regardless of the specific packing material utilized, the UV 280 nm chromatograms displayed three principal peaks, and complete baseline separation was not achieved between these peaks. Following the current process, only the first peak was collected for subsequent vaccine production, and the quality parameters were subsequently tested.

### 3.2. Analysis of First Peak Areas

In the early stage of process development, fraction collection was conducted to determine that the virus was mainly concentrated in the first peak. Therefore, this experiment only analyzed the first peak of SEC. In the initial analysis, the peak areas of the virus in the UV 280 nm chromatograms were quantified and categorized statistically according to the batches of the hydrolytic concentrated solution. Figure 2 illustrates the peak areas of the virus peaks in the UV 280 nm chromatograms during the purification of three batches of hydrolytic concentrated solution using four SEC packing materials. The average peak areas for batches A, B, and C were 2608.48, 2165.90, and 1779.90, respectively. Notably, these average peak areas exhibited statistically significant differences (*p* < 0.001). This initial analysis underscores that different batches of the hydrolytic concentrated solution exert a significant impact on the purification efficiency. Consequently, these variations among the different batches of hydrolytic concentrated solution should be taken into account for subsequent statistical analyses.

Secondly, the peak areas of the virus peaks were grouped and statistically analyzed based on the chromatography medium used during purification: Chromstar 6FF, Singarose 6FF, Bestarose 6B, and Focurose 6FF. The average peak areas corresponding to virus peaks during purification with these media were 2139.59, 2287.41, 2199.10, and 2139.59, respectively, and they showed no statistically significant differences (*p* = 0.91).

The size of peak areas is commonly used as an indicator of substance content. Consequently, it can be initially inferred that there are no significant statistical differences in the total protein content obtained during SEC purification when using the four packing materials. However, it is important to acknowledge that relying solely on peak area might not sufficiently determine the effectiveness of purification due to the presence of various impurities in the hydrolytic concentrated solution. Therefore, further analysis and verification are necessary.

To comprehensively assess the purification effectiveness, we conducted tests on the initial purification liquid obtained through SEC purification. These tests evaluated parameters such as total protein content, antigen content, Vero DNA residue, Vero HCP residue, and BSA residue. The results were categorized based on the batches of hydrolytic concentrated solution used. Figure 3 presents the analysis of quality parameters in the initial purification liquids. Our test results indicated statistical differences among the initial purification liquids obtained from three different batches of hydrolytic concentrated solution using the current process. Specifically, there were notable variations in terms of total protein content (*p* = 0.01), antigen content (*p* < 0.001), specific activity (*p* < 0.001), Vero HCP residue (*p* = 0.02), and Vero DNA residue (*p* < 0.001). These differences primarily occur due to variations among the different batches of hydrolytic concentrated solution utilized. By presenting these findings, we highlight the crucial role played by the batch of hydrolytic concentrated solution in affecting the purification effectiveness. Further analysis and verification are imperative to ascertain the specific factors contributing to these differences and to ensure consistent and reliable purification outcomes.

The most crucial parameters to consider during the SEC stage are antigen recovery and BSA residue. The results demonstrate that there are no statistically significant differences in antigen recovery (*p* = 0.89) and BSA residue (*p* = 0.28). Antigen recovery is closely associated with yield, while BSA residue represents the removal of large molecular weight impurities. Therefore, these parameters provide valuable insights into the purification process. On the other hand, Vero HCP is a highly heterogeneous mixture of proteins and can interact with Vero DNA. Due to its complexity, it is not typically used as an evaluation parameter for the SEC stage. Similarly, Vero DNA tends to adsorb onto the virus surface during ultrafiltration concentration, making it challenging to remove during SEC purification. Hence, it is not considered an assessment parameter for this stage. Overall, the current process exhibits good robustness during the SEC purification stage.

To assess the effectiveness of different packing materials, we grouped the results based on the materials utilized during SEC purification. Figure 4 presents the analysis of quality parameters in the initial purification liquids obtained after SEC purification of three batches of hydrolytic concentrated solution. The outcomes indicate no statistically significant differences in total protein content (*p* = 0.52), antigen content (*p* = 0.32), specific activity (*p* = 0.37), Vero DNA residue (*p* = 0.78), and BSA residue (*p* = 0.12) among the initial purification liquids obtained using different materials. However, it is worth noting that there are significant statistical differences in Vero HCP residue (*p* < 0.001) and antigen recovery (*p* < 0.001). Although Vero HCP residue is not a critical quality parameter, the variations in antigen recovery rates suggest substantial differences in the purification effectiveness achieved with different packing materials. Further investigation is required to understand the underlying factors contributing to these variations and make informed decisions regarding the optimal choice of packing material for SEC purification.

### 3.3. Virus Purification by Anion Exchange Chromatography

Further purification of the solution obtained by SEC purification was carried out using four different AEC media produced by different companies: Maxtar Q, Q Singarose, Diamond Q, and Q Focurose. These materials, which possess quaternary ammonium ligands, are known for their ability to effectively remove residual DNA. Table 3 provides an overview of the properties of these AEC materials. During all experiments, UV 280 nm was continuously monitored in real-time. Figure 5 displays the UV 280 nm chromatograms of three batches of hydrolytic concentrated solution purified using the four AEC materials. To be consistent with the formal production process in the workshop, a 0.5 M NaCl solution was used as the elution buffer. According to the methodology, samples were collected with specific start and end times as marked in the figures. However, for some experiments, the chromatograms appeared disordered, possibly due to poor rigidity of the medium, leading to instability.

The UV 280 nm spectra of the AEC resins reveal notable differences that significantly impact the efficiency of purification. The spectra for Maxtar Q and Diamond Q exhibit disorder and inconsistency in peak times. Q Focurose displays irregular spectra in certain experiments. The peak profile of Q Singarose is relatively more regular, with consistent retention times across all groups. However, further analysis and testing results are necessary to corroborate these observations.

To further evaluate the purification efficiency, we assessed the total protein content, antigen content, Vero DNA residue, Vero HCP residue, and BSA residue in the final pure solution obtained through AEC purification. The data were analyzed based on the batches of hydrolytic concentrated solution. Figure 6 presents the analysis of relevant quality parameters in the final pure solution. Total protein content was not displayed in the results as it fell below the lower detection limit. However, for the remaining parameters, there were no statistically significant differences observed in the final pure solution obtained through the current AEC purification process, except for antigen content (*p* < 0.001), which exhibited statistical differences due to batch variations. The results indicate that the current AEC purification stage of the process demonstrates good robustness in terms of antigen recovery (*p* = 0.10), Vero HCP residue (*p* = 0.59), Vero DNA residue (*p* = 0.28), and BSA residue (*p* = 0.97). These findings suggest that the AEC purification stage of the current process is consistent and reliable, exhibiting favorable performance across various batches.

To assess the purification effectiveness of different resins, we categorized the data based on the resins used during AEC purification. Figure 7 provides an analysis of relevant quality parameters in the final pure solution. The test results indicate statistically significant differences in Vero HCP residue (*p* < 0.001) and BSA residue (*p* < 0.001) in the final pure solution obtained when different resins were employed for purifying the initial crude solution. However, no statistically significant differences were observed in terms of antigen content (*p* = 0.98), antigen recovery (*p* = 0.98), and Vero DNA residue (*p* = 0.06). As mentioned earlier, Vero DNA residue is a critical parameter for evaluating AEC purification efficiency as the primary objective is to remove DNA residues. The absence of significant differences in Vero DNA residue suggests that all the different resins utilized in this study are equally effective at removing DNA residues. While there are statistical differences in BSA residue and Vero HCP residue, it is important to note that most protein impurities have already been eliminated during SEC purification. The residual levels of these impurities after AEC purification meet the relevant standards, indicating that all the resins used in this process effectively remove protein impurities. In summary, the findings indicate that different resins exhibit varying performance in terms of Vero HCP residue and BSA residue. However, considering that the overall purification goals have been achieved, with low levels of impurities in the final pure solution meeting the required standards, it can be concluded that the current AEC purification process is effective and meets the desired criteria for purification.

### 3.4. Correlation Analysis of Relevant Quality Parameters and Process Parameters

When analyzing the vaccine’s downstream production process, it is essential to consider numerous process conditions and evaluate their interactions comprehensively. Subsequently, we conducted a correlation analysis to identify any relationships between these factors.

The Figure 8 vividly illustrates the correlations between various process parameters and quality parameters, aiding in the identification of crucial process parameters associated with quality. In the SEC results, we noted specific correlations between different factors. Particularly, we observed that the cell density of the virus harvest displayed negative correlations with both antigen content (R² = −0.89) and the area of the virus peak in the UV 280 nm spectra (R² = −0.65). Additionally, column efficiency exhibited a positive correlation with antigen recovery rate (R² = 0.58), while asymmetry demonstrated a negative correlation with antigen recovery rate (R² = −0.56). However, in the AEC correlation analysis, no significant correlations with process conditions related to antigen content or antigen recovery rate were observed.

## 4. Discussion

In this study, we conducted an evaluation of the downstream production process for the current inactivated COVID-19 vaccine. Our aim was to assess the robustness and reliability of this process. To achieve this, we employed four gel filtration chromatography media and four anion exchange chromatography media to purify three batches of hydrolytic concentrated solution. We then tested various quality parameters in both the intermediate products and the original solution. Upon conducting a statistical analysis of the experimental results, we found that the current downstream process exhibited robustness and reliability. It consistently maintained high antigen recovery rates and complied with purity standards outlined in the pharmacopoeia, even when different batches of hydrolytic concentrated solution or different chromatography resins were employed.

Cell density can have various effects on antigen content within a certain range. Higher cell density is generally believed to enhance antigen yield because it provides more host cells for virus infection and replication. However, excessively high cell density can lead to overcrowding and limited nutrient availability, which can result in reduced virus production and subsequently affect antigen production [22]. Furthermore, cell density can also impact purification efficiency. Higher cell density can introduce more impurities, making the purification process more complex [23]. On the other hand, lower cell density may simplify purification but might require multiple cultures to obtain a sufficient antigen yield. A study by Thomassen YE et al., which utilized Vero cell cultures for poliovirus, demonstrated that increasing cell density could lead to a threefold increase in D-antigen levels [24]. However, when cells were infected at high densities, the yield of D-antigens per cell was lower. The negative correlation observed in our study between cell density and antigen content is likely due to nutrient limitations and cellular competition arising from excessively high cell density. These factors ultimately contribute to a reduction in antigen production. It is important to strike a balance in cell density to optimize antigen yield and ensure efficient purification processes.

The HETP of the chromatography column is an important factor in the purification process. It measures the column’s ability to separate target compounds [25]. The level of HETP directly impacts the quality of separation achieved. Columns with higher HETP values offer better resolution, making the distinctions between target compounds and impurities more pronounced. This is particularly crucial in industries such as pharmaceuticals and vaccines, where high-purity products are essential. In this study, it is expected that there would be a positive correlation between HETP and antigen recovery rate. As HETP increases, the column’s ability to retain and separate antigens improves, leading to higher antigen recovery rates. Another critical factor is asymmetry, which relates to the uniformity of the column’s packing material and the consistency of sample flow. Good symmetry ensures the even distribution of the sample within the column, preventing uneven distribution in certain areas that could result in component loss or poor purification results [26]. Asymmetry values closer to 1 are generally considered favorable, indicating symmetrical and well-defined peak shapes [27]. This type of peak shape suggests minimal sample dispersion as it passes through the column, signifying appropriate separation conditions and excellent column performance. In our study, we found that lower asymmetry values were associated with higher antigen recovery rates, which aligns with expectations. The smallest observed asymmetry value in this experiment was 0.99, very close to 1, indicating good peak symmetry and suggesting optimal separation conditions for efficient antigen recovery.

There were no valuable findings in the correlation analysis results of AEC. This can be attributed to the complexity of the behavior of the initial purification fluid in AEC and the possibility that the process conditions included in our analysis may not be comprehensive enough to capture all relevant factors. Unlike the process of hydrolytic concentrated solution in SEC, the behavior of the initial purification fluid in AEC involves intricate adsorption and desorption processes. These behaviors are influenced by various factors such as ion exchange capacity, ionic strength, protein properties, counterion type, and surface ligands, among others [28]. The complexity of these interactions may have contributed to the absence of obvious correlations within the known process conditions. Further investigation is warranted to gain a better understanding of the adsorption and desorption behavior of the initial purification fluid in the AEC process and to identify potential factors that may influence it but were not considered in our analysis. This may require more extensive experimentation and analysis to explore these intricate interactions comprehensively. Therefore, for future research, it would be valuable to expand the range of process conditions and consider additional factors to gain a deeper understanding of the behavior of the initial purification fluid in AEC. By conducting further studies and considering a broader range of process conditions, we can enhance our understanding of the complex interactions involved in the AEC process. This will contribute to optimizing downstream purification processes, improving antigen recovery rates, and enhancing the overall quality of the final product.

## 5. Conclusions

In this study, our focus was on evaluating the robustness and efficiency of the current downstream purification process for the inactivated COVID-19 vaccine. To accomplish this, we utilized four gel filtration chromatography media and four ion exchange chromatography media to purify the hydrolytic concentrated solution from three different batches. We examined various quality parameters and conducted statistical analysis to assess the purification efficiencies of these chromatography media. Based on our findings, we can draw several conclusions from this study. Firstly, we determined that the current downstream purification process exhibits satisfactory stability during both the gel filtration and ion exchange chromatography stages. This indicates that the process is robust and reliable for purification purposes. Among the four gel filtration chromatography media used, we found no statistically significant differences in purification efficiency. This suggests that all of these media are equally effective in purifying the hydrolytic concentrated solution. On the other hand, we observed statistically significant differences in purification efficiency among the four ion exchange chromatography media. This highlights the importance of carefully selecting the most appropriate media for production purposes. By choosing the optimal ion exchange chromatography media, we can enhance the purification efficiency and overall quality of the final vaccine product. Furthermore, our study suggests that this downstream production process holds promise for extending to the production of other virus-based vaccines. This opens up extensive application prospects in the field of vaccine manufacturing. Overall, our research confirms the stability and reliability of the current downstream purification process for the inactivated COVID-19 vaccine. It also emphasizes the significance of media selection in optimizing purification efficiency. Moreover, these findings pave the way for potential applications in the production of other virus-based vaccines, indicating a promising future for this downstream production process.

## Figures and Tables

**Figure 1 vaccines-12-00056-f001:**
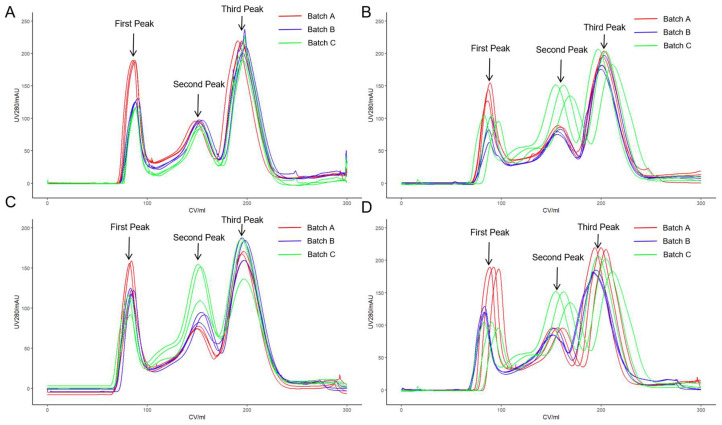
Purification of three batches of the hydrolytic concentrated solution by SEC chromatography. Different media were used, respectively: (**A**) Chromstar 6FF; (**B**) Singarose 6FF; (**C**) Bestarose 6B; (**D**) Focurose 6FF. The column specifications were XK 16/100, with PBS as the mobile phase at a flow rate of 1 mL/min.

**Figure 2 vaccines-12-00056-f002:**
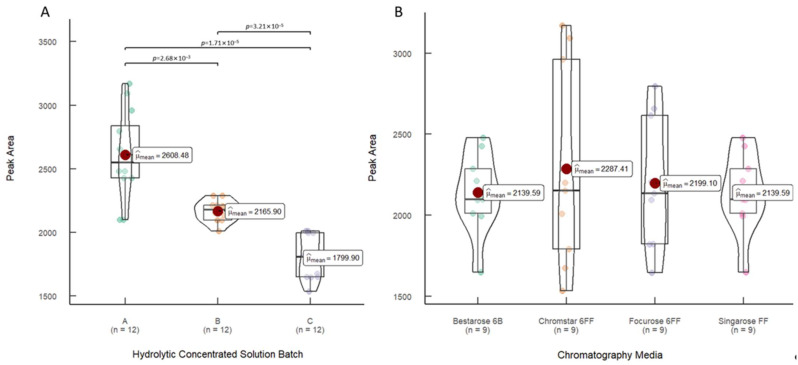
The peak areas of the virus peaks in the UV280 nm chromatograms. Statistics were conducted according to different grouping methods: (**A**) according to the batch of hydrolytic concentrated solution used; (**B**) according to the chromatography medium used.

**Figure 3 vaccines-12-00056-f003:**
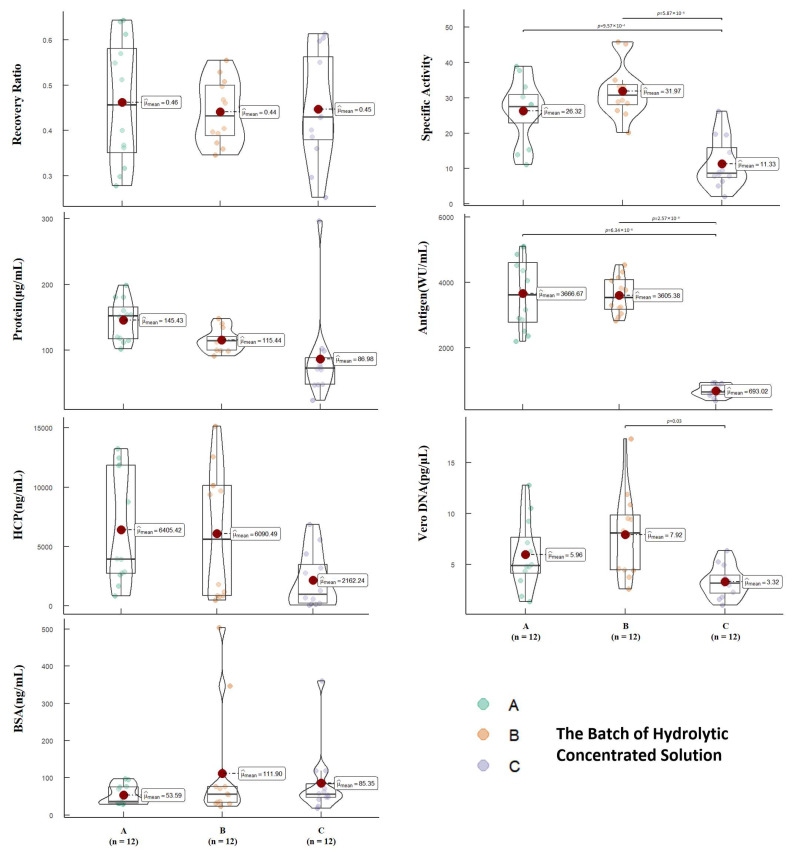
The impact of different batches of hydrolytic concentrated solution on the quality parameters, showing the effects of the batch on the antigen recovery rate, specific activity, protein content, antigen content, HCP content, Vero DNA content, and BSA content in the initial purified liquid obtained after SEC. Each point represents the results of the initial pure liquid detected in a set of experiments, and the unbiased estimate of the overall mean ($\hat{\mu }$_mean_) is also indicated on the graph.

**Figure 4 vaccines-12-00056-f004:**
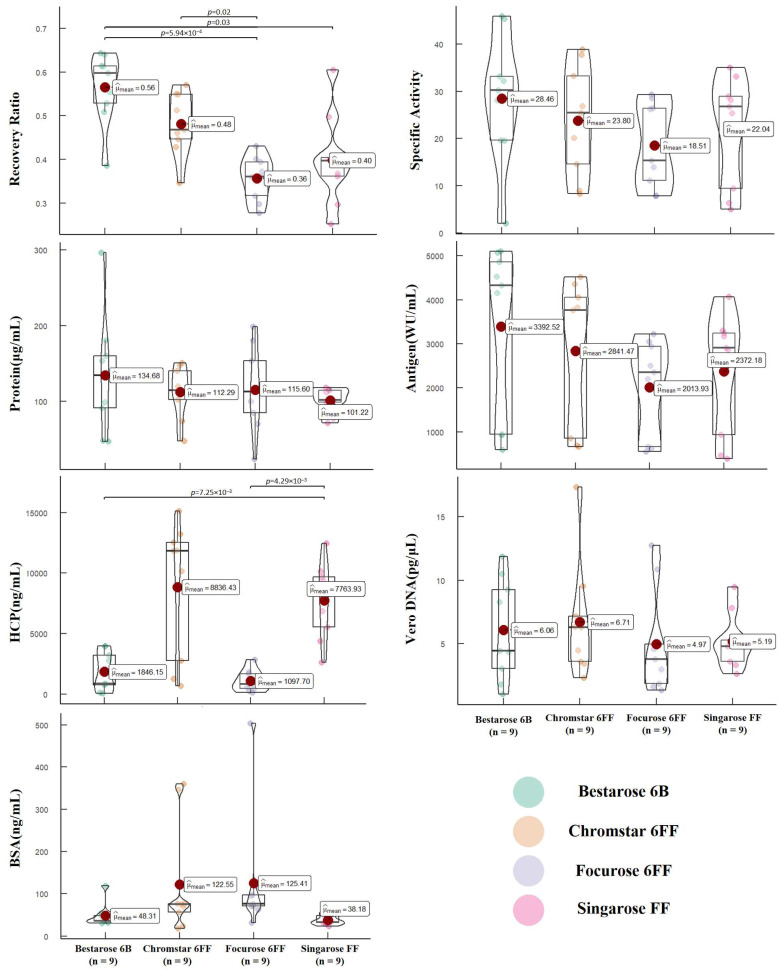
The impact of different chromatography medias on the quality parameters, showing the effects of four different chromatography medias on the antigen recovery rate, specific activity, protein content, antigen content, HCP content, Vero DNA content, and BSA content in the initial purified liquid obtained after SEC. Each point represents the results of the initial pure liquid detected in a set of experiments, and the unbiased estimate of the overall mean ($\hat{\mu }$_mean_) is also indicated on the graph.

**Figure 5 vaccines-12-00056-f005:**
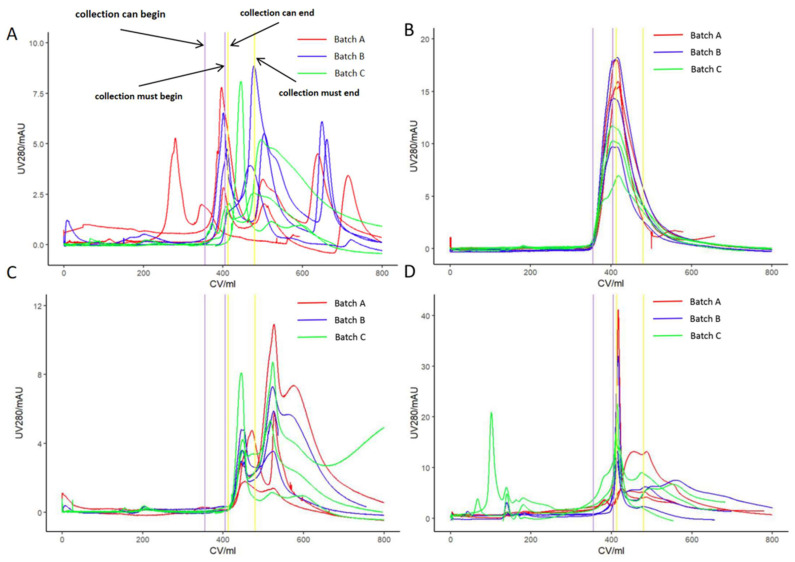
Purification of the initial purification liquids obtained through SEC purification by AEC chromatography. Different media were used, respectively: (**A**) Maxtar Q; (**B**) Q Singarose; (**C**) Diamond Q; (**D**) Q Focurose. The column specifications were XK 50/30, with 0.5 M NaCl as the elution buffer at a flow rate of 5 mL/min. Two purple vertical lines indicate the range where sample collection can and must begin (0.72–0.82 CV), while two yellow vertical lines indicate the range where sample collection can and must end (0.84–0.97 CV).

**Figure 6 vaccines-12-00056-f006:**
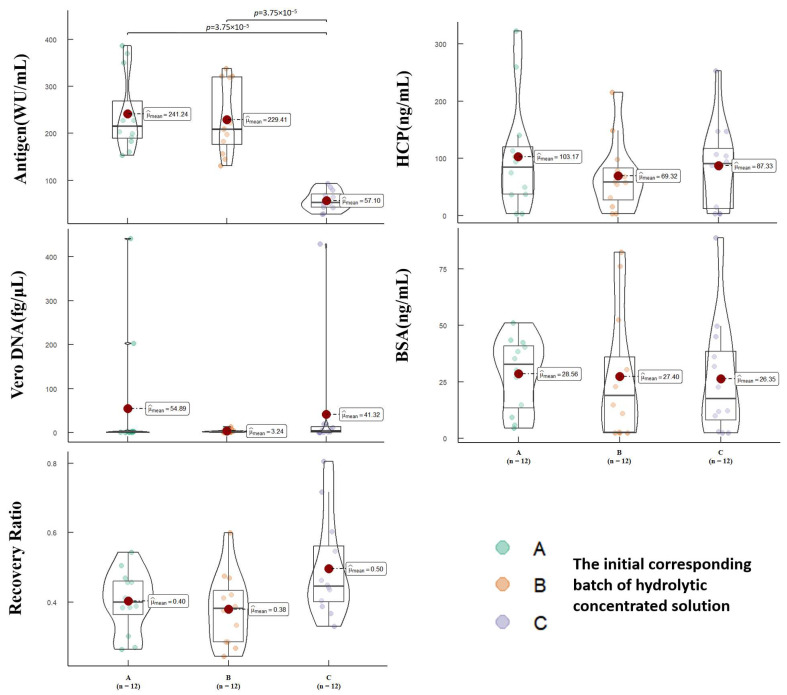
The impact of different batches of hydrolytic concentrated solution on the quality parameters, showing the effects of the batch on the antigen recovery rate, specific activity, protein content, antigen content, HCP content, Vero DNA content, and BSA content in the final purified liquid obtained after AEC. Each point represents the results of the final purified liquid detected in a set of experiments, and the unbiased estimate of the overall mean ($\hat{\mu }$_mean_) is also indicated on the graph.

**Figure 7 vaccines-12-00056-f007:**
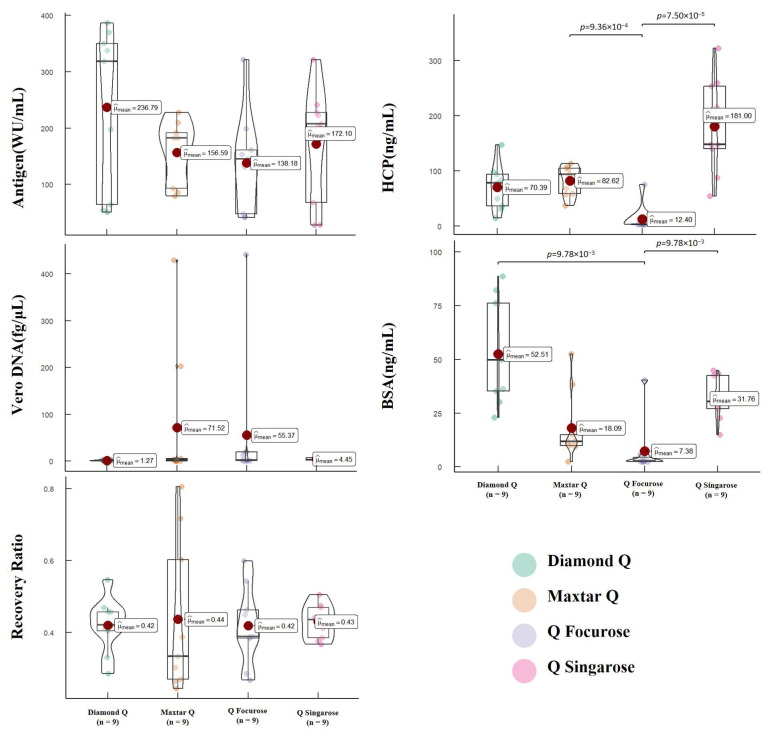
The impact of different AEC chromatography medium on the quality parameters, showing the effects of different AEC chromatography medias on the antigen recovery rate, specific activity, protein content, antigen content, HCP content, Vero DNA content, and BSA content in the final purified liquid obtained after AEC. Each point represents the results of the final purified liquid detected in a set of experiments, and the unbiased estimate of the overall mean ($\hat{\mu }$_mean_) is also indicated on the graph.

**Figure 8 vaccines-12-00056-f008:**
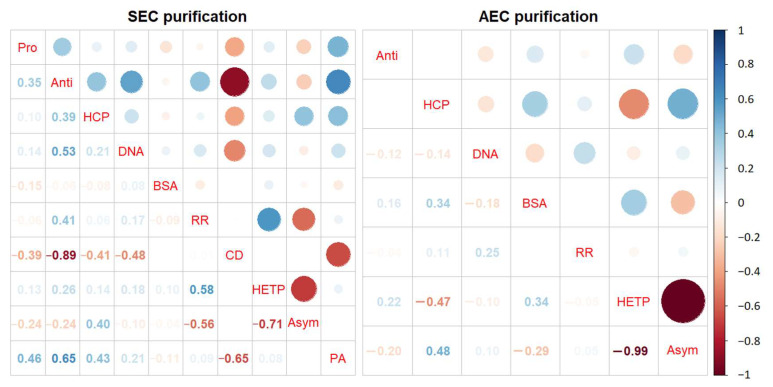
Correlation Analysis Matrix Figure. The left panel shows the correlation analysis between the quality parameters of the samples after SEC purification, the SEC column-related parameters, and cell culture data. The right panel displays the correlation analysis between the quality parameters of the samples after AEC purification, the AEC column-related parameters, and cell culture data. The abbreviations on the diagonal are as follows: Pro: Protein; Anti: Antigen; HCP: Vero Host Cell Protein; DNA: Vero DNA; BSA: Bovine Albumin; RR: Antigen Recovery Rate; CD: Cell Density; HETP: Height Equivalent of Theoretical Plate; Asym: Asymmetry; PA: Peak Area. In the figure, red represents negative correlation, blue represents positive correlation, and the darker the color, the greater the correlation coefficient.

**Table 1 vaccines-12-00056-t001:** Upstream operational conditions and quantification.

NO.	Strain	Bioreactor Volume	Cell Density at Infection	MOI	Trypsin	Temperature at Infection	Time of Harvest	Total Cell Density at Harvest
Run#1	Omicron B.1.1.529	1000 L	3.54 × 10^6^ cells/mL	0.0001	1 μg/mL	37 °C	113 hpi	2.655 × 10^6^ cells/mL
Run#2	Omicron B.1.1.529	1000 L	3.42 × 10^6^ cells/mL	0.0001	1 μg/mL	37 °C	115 hpi	2.565 × 10^6^ cells/mL
Run#3	Omicron B.1.1.529	1000 L	4.6 × 10^6^ cells/mL	0.0001	1 μg/mL	37 °C	110 hpi	3.45 × 10^6^ cells/mL

**Table 2 vaccines-12-00056-t002:** Specifications of SEC chromatography columns.

#	Medium	Exclusion Limit	Bed Height (cm)	Volume (mL)	HETP	Asym
1	Chromstar 6FF	10~4000 kDa	85	170.9	2976	1.17
2	Singarose 6FF	10~4000 kDa	86	172.9	2599	1.39
3	Bestarose 6B	10~4000 kDa	85	170.9	2891	0.99
4	Focurose 6FF	10~4000 kDa	85	170.9	2647	1.20

**Table 3 vaccines-12-00056-t003:** Specifications of AEC chromatography columns.

#	Medium	Exclusion Limit	Bed Height (cm)	Volume (mL)	HETP	Asym
1	Maxtar Q	10~4000 kDa	25	490.87	2822	1.23
2	Q Singarose	10~4000 kDa	25	490.87	2977	1.22
3	Diamond Q	10~4000 kDa	25	490.87	2608	0.8
4	Q Focurose	10~4000 kDa	25	490.87	2633	0.94

## Data Availability

The data presented in this study are available upon request from the corresponding author.

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
