# Peer review of "Evaluation of the Robustness Verification of Downstream Production Process for Inactivated SARS-CoV-2 Vaccine and Different Chromatography Medium Purification Effects"

_vaccines, 2024, doi:10.3390/vaccines12010056_

Round 1
Reviewer 1 Report
Comments and Suggestions for Authors
In this paper, the authors evaluated the robustness of their purification process to produce inactivated SARS-CoV-2 vaccine with different batch of feeds and with resins from different manufacturers, and the results showed the robustness is there. However, this study is just a set of routine experiments with standard process and lacks novelty.
Author Response
Please see the attachment.
For research article
Response to Reviewer 1 Comments
|
||
1. Summary |
|
|
Thank you for your time and effort in reviewing this manuscript. Below, you will find our detailed responses to your comments, along with the corresponding revisions and corrections clearly highlighted or tracked in the resubmitted files. We greatly appreciate your thorough evaluation and valuable feedback.
|
||
2. Questions for General Evaluation |
Reviewer’s Evaluation |
Response and Revisions |
Does the introduction provide sufficient background and include all relevant references?
|
Yes |
|
Are all the cited references relevant to the research?
|
Yes |
|
Is the research design appropriate?
|
Yes |
|
Are the methods adequately described? |
Yes
|
|
Are the results clearly presented? |
Yes
|
|
Are the conclusions supported by the results? |
Yes
|
|
3. Point-by-point response to Comments and Suggestions for Authors |
||
Comments 1: In this paper, the authors evaluated the robustness of their purification process to produce inactivated SARS-CoV-2 vaccine with different batch of feeds and with resins from different manufacturers, and the results showed the robustness is there. However, this study is just a set of routine experiments with standard process and lacks novelty.
|
||
Response 1: Thank you very much for your review of our paper and for your valuable feedback. We fully understand your concerns regarding the level of innovation in our study. We would like to clarify the innovative aspects of our research:
1.In-depth Detail: While our study adopts standard manufacturing processes, we conducted in-depth investigations using virus concentrates from different batches and chromatography resins from various manufacturers. This contributes to filling the gaps in detailed research within the relevant field.
2.Robustness Analysis: Our study primarily evaluates the robustness of the current purification process in producing inactivated SARS-CoV-2 B.1.1.529 strainstrain vaccines. This is crucial in ensuring the reliability of the vaccine production process, as an ideal production process should maintain consistency across batches when dealing with different raw materials and equipment.
3.Practical Application Significance: Our work addresses challenges that may arise in real-world production environments. Other researchers can utilize the results of our experiment to select appropriate resins or attempt to expand the application of this production process to purify other viruses. This holds practical significance for optimizing vaccine production processes and problem-solving.
In summary, we believe that while our study utilizes standard process flows, the in-depth exploration of details and robustness analysis contribute to its innovation and practical applicability.
|
||
4. Response to Comments on the Quality of English Language |
||
Point 1:English language fine. No issues detected |
||
Response 1: Thank you for your feedback regarding the quality of the English language. I appreciate your comment and I'm glad to hear that there were no issues detected. If you have any further questions or concerns, please feel free to let me know.
|
||
5. Additional clarifications |
||
I have reviewed the comments and suggestions provided by the journal editor and reviewer, and I would like to express my gratitude for their thorough review of my work. After careful consideration, I do not have any additional clarifications to provide at this time. I believe that the manuscript adequately addresses the research questions and presents the findings in a clear and concise manner. However, I am open to further discussion or clarification if requested by the editor or reviewer. |

Reviewer 2 Report
Comments and Suggestions for Authors
These are my comments, some minor some major.
Line 91-93, separate units from dimensions, 20.0L/min must be written as 20.0 L/min. Do this throughout the document.
Line 95, use "mL" instead of "ml"
Section 2.2, how is the virus separated form the cells? The virus is dispersed in what?
Line 104-105, how was the inactivation verification step performed?
Section 2.3, during clarification, what is being separated from what? The virus is dispersed in what?
Line 113, change "cm2" to "cm2"
Line 115, change "KDa" to "kDa" and "m2" to "m2"
Line 126, what is the column size? (the bed volume). Did you pack the columns? How was this step performed?
Line 129, change "ml/min" to "mL/min"
What was the injection volume for SEC? Was regeneration applied between batches and/or injections?
Line 141, what is the column size? (the bed volume). Did you pack the columns? How was this step performed?
Line 144, PBS for anion exchange chromatography is a poor choice as phosphate will compete for the adsorption sites and the moderate salt concentration will also hinder the adsorption of DNA
Line 145, change "ml/min" to "mL/min"
What was the injection volume for AEC? Was regeneration applied between batches and/or injections?
Line 146-147, the virus is negatively charged then?
Section 2.6.5, more information must be provided with respect to antibodies and enzyme-linked antibodies.
Line 194, you are injecting too much sample, not complete resolution is expected
Line 195-196, "these three peaks were the virus, high molecular weight protein impurities, and medium molecular weight protein impurities", provide evidence of this.
Figure 1, there is another substance eluting at around 120 mL, was its area added to the second peak? Did you analyze a fraction closed to this elution volume?
Figure 3, all labels cannot be distinguished. The results presented here are for the first peak of Figure 1?
Figure 4, all labels cannot be distinguished
Line 283, 0.5 M NaCl, the methodology established a 1 M NaCl solution for desorption.
Line 284-285, "This elution buffer resulted in a high viral content and low impurities in the first elution peak", do a better description. What peak? What column?
Line 285-286, "Consequently, in this specific experiment, only the first elution peak was collected for subsequent analysis", for what chromatographic medium? Be specific.
Figure 5, the title of the chromatograph in the upper left says "Chromastar 6FF". Indicate with an arrow when the entrance of 0.5 M NaCl starts.
Line 294-296, "The spectra for Maxtar Q and Diamond Q exhibit disorder and inconsistency in peak times, along with the presence of numerous impurities", did you regenerate the column between injections and between batches?
Line 296-298, "Q Singarose demonstrates the best performance, exhibiting well-defined spectra with consistent peak times and no impurities", provide evidence of this.
Line 301-302, "in the final pure solution obtained through AEC purification", you need to be specific, "a sample collected from 400 to 450 mL of elution volume", something like that.
Figures 6 and 7, nothing can be read in these plots.
A lot of analyses are provided. Despite this, regular yield and purity must be calculated. What is the antigen yield? What are the initial and final purities of the antigen? The resulting antigen concentration is enough for immunization?
For AEX, it is not clear what peak did you compare between columns. A difficult task considering the chromatograms of Maxtar and Diamond.
Author Response
Please see the attachment.
For research article
Response to Reviewer 2 Comments
|
||||||||||||||||||||||||||||||||||||||||||||||||||||||||||||||||||||||||
1. Summary |
|
|
||||||||||||||||||||||||||||||||||||||||||||||||||||||||||||||||||||||
Thank you for your time and effort in reviewing this manuscript. Below, you will find our detailed responses to your comments, along with the corresponding revisions and corrections clearly highlighted or tracked in the resubmitted files. We greatly appreciate your thorough evaluation and valuable feedback.
|
||||||||||||||||||||||||||||||||||||||||||||||||||||||||||||||||||||||||
2. Questions for General Evaluation |
Reviewer’s Evaluation |
Response and Revisions |
||||||||||||||||||||||||||||||||||||||||||||||||||||||||||||||||||||||
Does the introduction provide sufficient background and include all relevant references? |
Yes |
|
||||||||||||||||||||||||||||||||||||||||||||||||||||||||||||||||||||||
Are all the cited references relevant to the research? |
Yes |
|
||||||||||||||||||||||||||||||||||||||||||||||||||||||||||||||||||||||
Is the research design appropriate? |
Can be improved |
We have provided our corresponding response in the point-by-point response letter. |
||||||||||||||||||||||||||||||||||||||||||||||||||||||||||||||||||||||
Are the methods adequately described? |
Can be improved |
We have provided our corresponding response in the point-by-point response letter. |
||||||||||||||||||||||||||||||||||||||||||||||||||||||||||||||||||||||
Are the results clearly presented? |
Must be improved |
We have provided our corresponding response in the point-by-point response letter. |
||||||||||||||||||||||||||||||||||||||||||||||||||||||||||||||||||||||
Are the conclusions supported by the results? |
Must be improved |
We have provided our corresponding response in the point-by-point response letter. |
||||||||||||||||||||||||||||||||||||||||||||||||||||||||||||||||||||||
3. Point-by-point response to Comments and Suggestions for Authors |
||||||||||||||||||||||||||||||||||||||||||||||||||||||||||||||||||||||||
Comments 1: Line 91-93, separate units from dimensions, 20.0L/min must be written as 20.0 L/min. Do this throughout the document. Line 95, use "mL" instead of "ml" Line 145, change "ml/min" to "mL/min" Line 113, change "cm2" to "cm2" Line 115, change "kDa" to "kDa" and "m2" to "m2" Line 129, change "ml/min" to "mL/min" |
||||||||||||||||||||||||||||||||||||||||||||||||||||||||||||||||||||||||
Response 1: Thank you for providing specific comments and suggestions for the formatting of our document. We have carefully reviewed your feedback and made the necessary revisions as per your instructions.
The the format have been implemented throughout the document.
We would like to express our gratitude for your meticulous review and assure you that all the recommended changes have been diligently addressed to ensure the accuracy and consistency of the document.
|
||||||||||||||||||||||||||||||||||||||||||||||||||||||||||||||||||||||||
Comments 2: Section 2.2, how is the virus separated form the cells? The virus is dispersed in what?
|
||||||||||||||||||||||||||||||||||||||||||||||||||||||||||||||||||||||||
Response 2: 1.The clinical samples used were bronchoalveolar lavage fluid obtained from patients, and these clinical samples originated from a hospital in Hong Kong. After obtaining the clinical samples, they were inoculated into Vero-E6 cells in the P3 laboratory for passage cultivation. Cell pathology was observed during the cultivation process. Cells showing pathological changes were selected for virus RNA extraction and sequencing.
2.Please rest assured that we strictly adhere to relevant biosafety regulations and laboratory operation standards to ensure compliance and safety in our virus isolation work. As this study falls within the scope of production technology, the isolation of the virus strain was only briefly mentioned in the manuscript without detailed description. We have now made revisions to the article, adding more detailed information to address your concerns. We apologize for the lack of detail in the previous description and appreciate your feedback.
“The Vero cells utilized in this experiment were originally cryopreserved by the newly established Infectious Disease Laboratory. The Omicron BA.1 strain of SARS-CoV-2 was isolated from clinical samples by the Infectious Disease Laboratory. ”(Lines 86-88) “The Vero cells used in this experiment were obtained from the newly emerging infectious diseases laboratory and were cryopreserved. The viral strain used was the Omicron B.1.1.529 strain of SARS-CoV-2, initially isolated from clinical samples (bronchoalveolar lavage fluid) from Hong Kong, and inoculated into Vero-E6 cells in the P3 laboratory.”(Lines 88-92)
Comments 3:Line 104-105, how was the inactivation verification step performed?
Response 3:In our study, the inactivation validation employed a series of standardized experimental methods, including virus inactivation efficacy determination and virus replication detection in cell culture, to comprehensively assess the effectiveness and safety of virus inactivation. After each inactivation, we took a portion of the inactivated samples for continuous cultivation for 3 generations in Vero cells, during which cell pathology was observed, and RNA was extracted for sequencing to confirm complete inactivation.
These methods are commonly used standard operating procedures in the industry, ensuring the appropriate concentration of the inactivating agent and sterilization time.
Simultaneously, we strictly adhered to relevant biosafety regulations and laboratory operation standards to ensure that all experimental procedures were compliant and conducted in a safe environment. We meticulously documented the process of inactivation validation and rigorously controlled experimental conditions to ensure the reliability and scientific integrity of the results.
We have now made revisions to the article, adding more detailed information to address your concerns. We apologize for the lack of detail in the previous description and appreciate your feedback.
“Upon completion of the inactivation step, a portion of the sample was extracted for inactivation verification to ensure the thorough inactivation process had been successfully achieved before proceeding to the subsequent steps.”(Lines 109-111) “After the inactivation is completed, a portion of the samples should be extracted for inactivation verification. This involves transferring the inactivated samples into Vero cells and culturing them for three consecutive generations. The purpose is to observe any signs of pathology and conduct sequencing to ensure that complete inactivation has been successfully achieved before proceeding with subsequent steps.”(Lines 112-116)
Comments 4:Section 2.3, during clarification, what is being separated from what? The virus is dispersed in what?
Response 4:We appreciate your insightful query regarding Section 2.3 of the manuscript, and we would like to provide clarification on the components being separated during the specified step.
During the cultivation process, due to the adherent growth characteristics of Vero cells, we employed spherical microcarriers to facilitate the growth of the cells. In this particular step, the microcarriers can be effectively removed from the culture medium. Additionally, this process enables the removal of cellular debris, cell membrane remnants, and large molecular substances present in the serum used during the cultivation.
It is important to note that during the growth of SARS-CoV-2 virus in Vero cells, the virus continually proliferates and buds, leading to the lysis of the host cells. As a result, the majority of the virus is present in the supernatant fraction of the harvested liquid.
We hope that this explanation provides clarity on the specific components being separated and the context in which the separation occurs. Thank you for bringing attention to this point, and we believe that this information will enhance the understanding of the described process.
We welcome any further inquiries or feedback and are committed to ensuring the accuracy and comprehensiveness of our manuscript.
Comments 5:Line 126, what is the column size? (the bed volume). Did you pack the columns? How was this step performed? Line 141, what is the column size? (the bed volume). Did you pack the columns? How was this step performed?
Response 5:1.Specifications and bed heights of the SEC and AEC chromatography columns are both mentioned. SEC : "The column specifications were XK 16/100, with PBS as the mobile phase at a flow rate of 1 mL/min."(Line 258) AEC : "The column specifications were XK 50/30"(Lines 364-365) The volumes and heights of the SEC and AEC columns are shown in Table 2 (Line 252) and Table 3 (Line 360). Table 2. Specifications of SEC chromatography columns.(Line 252 )
Table 3. Specifications of AEC chromatography columns.(Line 360)
2.During the packing process, I followed the instructions to ensure compaction and measured column efficiency and symmetry. Only after meeting the criteria did I proceed to the next step. I have made some modifications to my previous description. "Once column efficiency was confirmed, the subsequent steps of the experiment were initiated." (Line 132) "Once column efficiency exceeded 2500 and symmetry fell within the range of 0.8 to 1.5, the subsequent steps of the experiment were initiated."(Lines 137-139)
”Prior to proceeding to the subsequent steps, the columns were evaluated for performance.”(Line154) ”At this point, consistent with the approach taken with the SEC, it is necessary to test column efficiency and symmetry before proceeding with subsequent steps.”(Lines 166-168)
Comments 6:What was the injection volume for SEC? Was regeneration applied between batches and/or injections?
Response 6:It was unclear in my description, and I have made revisions.
”Subsequently, the concentrated hydrolytic solution was introduced into the chromatography column, and the virus was isolated from other impurities using the same buffer and flow rate. The constituents of all eluates were monitored by measuring the UV absorbance at 280 nm, and the virus fraction was collected. This procedure was conducted in triplicate for each type of gel filtration medium.”(Line 135)
”The sample is the liquid collected from SEC, and the sample volume at this time is 30 mL, approximately 6.1% of the CV. In the subsequent process, the flow rate should be set to 5 mL/min. Immediately after loading the sample, switch the liquid to 0.5 M NaCl. The specific method for collecting samples is as follows, in accordance with the current practices in the workshop. The collection of samples can begin at 0.72 CV, which means no substances should be collected before this point, even if there is material detected at UV 280 nm. The collection must start at 0.82 CV, even if no material is detected at UV 280 nm at this time. The collection of samples can be completed at 0.84 CV, which means it must continue until this point, even if no material flows out at UV 280 nm before this. The collection of samples must be completed at 0.97 CV, even if there is still material flowing out at UV 280 nm at this time.Regenerate between each set of experiments using 5 CV of 1 M NaOH solution, followed by washing with 5 CV of purified water, and finally equilibrating with at least 5 CV of PBS buffer until UV 280 nm and conductivity values stabilize to ensure the independence of each set of experiments. Repeat the entire process three times for each chromatographic medium.”(Lines 173-187)
Comments 7:Line 144, PBS for anion exchange chromatography is a poor choice as phosphate will compete for the adsorption sites and the moderate salt concentration will also hinder the adsorption of DNA
Response 7:Thank you for providing valuable feedback on our study regarding the use of PBS for anion exchange chromatography. We appreciate your insights and would like to address your concerns.
While we acknowledge that phosphate in PBS may compete for adsorption sites, and the moderate salt concentration could potentially hinder DNA adsorption during the purification process, it is important to note that PBS is a commonly used buffer for purifying inactivated vaccines. Additionally, PBS can effectively maintain antigen stability during and after the purification process. Our laboratory has extensive experience and well-established compatibility of PBS with many vaccine(EVA-71,CVA-10,CVA-6) purification processes, which is why we continue to use it.
Regarding the impact on DNA removal, we have conducted thorough assessments to ensure that residual DNA levels meet the required standards for vaccine production. Although PBS may influence DNA removal to some extent, the final residual DNA levels are within acceptable limits and comply with regulatory guidelines.
We appreciate your suggestion to explore alternative buffer systems in future experiments, and we will take it into consideration. We plan to investigate the application of different buffers to optimize the purification process further.
Once again, we thank you for your constructive feedback, and we assure you that your comments have been duly noted and will be taken into account as we continue to improve our experimental procedures.
Comments 8:What was the injection volume for AEC? Was regeneration applied between batches and/or injections?
Response 8:It was unclear in my description, and I have made revisions.
”The columns underwent an initial wash with purified water for five column volumes(CV). Subsequently, they were equilibrated with PBS buffer adjusted to a pH of 7.4 at a flow rate of 5 mL /min. The initial purified samples from company-specific gel filtration chromatography were then introduced into the respective AEC columns. Elution was carried out using 1 M NaCl, and the virus fraction was collected. This entire process was repeated in triplicate for each type of chromatography medium. ”(Line148)
”The sample is the liquid collected from SEC, and the sample volume at this time is 30 mL, approximately 6.1% of the CV. In the subsequent process, the flow rate should be set to 5 mL/min. Immediately after loading the sample, switch the liquid to 0.5 M NaCl. The specific method for collecting samples is as follows, in accordance with the current practices in the workshop. The collection of samples can begin at 0.72 CV, which means no substances should be collected before this point, even if there is material detected at UV 280 nm. The collection must start at 0.82 CV, even if no material is detected at UV 280 nm at this time. The collection of samples can be completed at 0.84 CV, which means it must continue until this point, even if no material flows out at UV 280 nm before this. The collection of samples must be completed at 0.97 CV, even if there is still material flowing out at UV 280 nm at this time.Regenerate between each set of experiments using 5 CV of 1 M NaOH solution, followed by washing with 5 CV of purified water, and finally equilibrating with at least 5 CV of PBS buffer until UV 280 nm and conductivity values stabilize to ensure the independence of each set of experiments. Repeat the entire process three times for each chromatographic medium.”(Lines 173-187)
Comments 9:Line 146-147, the virus is negatively charged then?
Response 9:Thank you for your question regarding the charge of the virus mentioned in our manuscript. Yes, the virus carries a negative charge, which enables it to adsorb onto the chromatographic medium. Subsequently, we employ 0.5M NaCl to desorb the virus.
We hope this clarification addresses your query, and we appreciate your attention to this detail.
Comments 10:Section 2.6.5, more information must be provided with respect to antibodies and enzyme-linked antibodies.
Response 10:Thank you for your feedback regarding Section 2.6.5 of our manuscript. We have now supplemented the section with additional information concerning the antibodies and enzyme-linked antibodies as per your suggestion.
“Antigen content was measured using the ELISA method. The standard for the novel coronavirus antigen content was prepared and stored in our institution's quality assurance laboratory. Antibodies and enzyme-linked antibodies were prepared by the virus vaccine research department of our institution. All subsequent operations were conducted in accordance with our institution's "SOP for Novel Coronavirus Inactivated Vaccine (Vero Cell) Antigen Content Detection.”(Line 175) “The ELISA method was used to measure the antigen content. The standard for the novel coronavirus antigen content was prepared and stored in our institution's quality assurance laboratory. Antibodies were prepared using serum from rabbits immunized with the novel coronavirus, and HRP Goat Anti-Rabbit IgG (#BA1054, Boster Biological Technology Co. Ltd, Wuhan, China) was used in the experiment. All subsequent operations were conducted according to our institution's SOP for Novel Coronavirus Inactivated Vaccine (Vero Cell) Antigen Content Detection.The novel coronavirus antigen content standard and samples were diluted to the initial dilution of 200 µL/well, and added to the coated plate in duplicate. Two-fold serial dilutions of not less than 6 dilutions were made in each well. The plate was placed in a constant temperature water bath at 37°C for 1 hour. The plate was washed 5 times with PBST washing solution. The detection antibody was added at the working concentration of 100 µL/well and incubated in a constant temperature water bath at 37°C for 0.5 hours. The plate was washed 5 times again. The enzyme-labeled antibody was added at the working concentration of 100 µL/well and incubated in a constant temperature water bath at 37°C for 0.5 hours. The plate was washed again 5 times. TMB coloration solution was added at 100 µL/well, mixed well, and then placed in a 37°C water bath for 15 minutes. After adding the stop solution at 50 µL/well, the OD value was measured at 450/630 nm, and all sample OD values were subtracted from the average OD value of the negative control. The antigen content of the test sample was calculated using the double parallel line method.”(Lines 208-228)
We trust that the revised section adequately addresses your concerns, and we appreciate your valuable input.
Comments 11:Line 194, you are injecting too much sample, not complete resolution is expected
Response 11:Thank you for your comments on our paper. We fully understand your point that reducing the sample injection volume indeed increases resolution. However, our experiment aims to simulate real-life scenarios in production. In large-scale production, we must consider efficiency and quality comprehensively. According to our experimental conditions, the chromatographic column used in production is of the scale 300/750, with a CV typically ranging from 35.3 to 56.5L. To meet production requirements, at times, the sample injection volume may reach up to 25%CV. Nevertheless, through appropriate subsequent purification steps, we can ensure that the final results meet all vaccine standards.
We hope our explanation provides a clearer understanding of our experimental design. Thank you for reviewing our paper again.
Comments 12:Line 195-196, "these three peaks were the virus, high molecular weight protein impurities, and medium molecular weight protein impurities", provide evidence of this.
Response 12:During the earlier stage of process development, we conducted segmented collection and comprehensive testing of each part, including antigen content, BSA residue, Vero host protein and host DNA residues, as well as total protein content. Additionally, we identified the purity of each collected fraction through SDS-PAGE and Western blot, thereby establishing the primary components of each peak. Ultimately, we confirmed the current process. Therefore, in this experiment, we did not perform segmented collection but rather collected only the first peak according to the current process.
We hope this information provides the evidence you requested. Thank you for reviewing our paper and providing valuable feedback.
Comments 13:Figure 1, there is another substance eluting at around 120 mL, was its area added to the second peak? Did you analyze a fraction closed to this elution volume? Response 13:We conducted segmented analysis during the earlier stage of process development, but in this experiment, we operated solely according to the established process and collected only the first peak for analysis.
We hope this addresses your question.
Comments 14:Figure 3, all labels cannot be distinguished. The results presented here are for the first peak of Figure 1? Figure 4, all labels cannot be distinguished Response 14:We have modified the figures, removing many of the small labels containing precise statistical results, which will be provided in an appendix for reference. The analysis of the results is focused only on the first peak collected. Figure 3. The impact of different batches of hydrolytic concentrated solution on the quality parameters. Show the effects of the batch on the antigen recovery rate, specific activity, protein content, antigen content, HCP content, Vero DNA content, and BSA content in the initial purified liquid obtained after SEC. Each point represents the results of the initial pure liquid detected in a set of experiments, and the unbiased estimate of the overall mean (mean)is also indicated on the graph.(Line 295)
Figure 4. The impact of different chromatography medias on the quality parameters. Show the effects of four different chromatography medias on the antigen recovery rate, specific activity, protein content, antigen content, HCP content, Vero DNA content, and BSA content in the initial purified liquid obtained after SEC.Each point represents the results of the initial pure liquid detected in a set of experiments, and the unbiased estimate of the overall mean (mean)is also indicated on the graph.(Line 342)
We hope this addresses the issues you raised. Thank you for reviewing our paper and providing valuable feedback.
Comments 15:Line 283, 0.5 M NaCl, the methodology established a 1 M NaCl solution for desorption.
Response 15:I'm sorry for the oversight, it should be a 0.5M NaCl solution for desorption. We have already updated the methodology section accordingly.
“Elution was carried out using 1 M NaCl, and the virus fraction was collected.”(Line 152) “In the subsequent process, the flow rate should be set to 5 mL/min. Immediately after loading the sample, switch the liquid to 0.5 M NaCl. “(Lines 174-175)
Thank you for reviewing and pointing this out in our paper.
Comments 16:Line 284-285, "This elution buffer resulted in a high viral content and low impurities in the first elution peak", do a better description. What peak? What column? Figure 5, the title of the chromatograph in the upper left says "Chromastar 6FF". Indicate with an arrow when the entrance of 0.5 M NaCl starts.
Response 16:We will modify the figures to label the start and end times of sample collection. Additionally, we have supplemented specific sampling steps for reference in the methodology section.
Figure 5. Purification of the initial purification liquids obtained through SEC purification by AEC chromatography. Different media were used respectively: (A) Maxtar Q; (B) Q Singarose; (C) Diamond Q; (D) Q Focurose. The column specifications were XK 50/30, with 0.5 M NaCl as the elution buffer at a flow rate of 5 mL/min. Two purple vertical lines indicate the range where sample collection can and must begin (0.72-0.82 CV), while two yellow vertical lines indicate the range where sample collection can and must end (0.84-0.97 CV).(Line 363)
Lastly, we have also made modifications to the description in this section.
“The specific method for collecting samples is as follows, in accordance with the current practices in the workshop. The collection of samples can begin at 0.72 CV, which means no substances should be collected before this point, even if there is material detected at UV 280 nm. The collection must start at 0.82 CV, even if no material is detected at UV 280 nm at this time. The collection of samples can be completed at 0.84 CV, which means it must continue until this point, even if no material flows out at UV 280 nm before this. The collection of samples must be completed at 0.97 CV, even if there is still material flowing out at UV 280 nm at this time.”(Lines 176-183)
We hope these changes address your concerns.
Comments 17:Line 285-286, "Consequently, in this specific experiment, only the first elution peak was collected for subsequent analysis", for what chromatographic medium? Be specific.
Response 17:I apologize for the previous unclear description. The specific method for sample collection has been revised as follows, in accordance with current practices in the workshop:
“The specific method for collecting samples is as follows, in accordance with the current practices in the workshop. The collection of samples can begin at 0.72 CV, which means no substances should be collected before this point, even if there is material detected at UV 280 nm. The collection must start at 0.82 CV, even if no material is detected at UV 280 nm at this time. The collection of samples can be completed at 0.84 CV, which means it must continue until this point, even if no material flows out at UV 280 nm before this. The collection of samples must be completed at 0.97 CV, even if there is still material flowing out at UV 280 nm at this time.”(Lines 176-183)
We appreciate your valuable feedback and suggestions.
Comments 18:Line 294-296, "The spectra for Maxtar Q and Diamond Q exhibit disorder and inconsistency in peak times, along with the presence of numerous impurities", did you regenerate the column between injections and between batches?
Response 18:We conducted column regeneration and made modifications and additions to the methodology.
2.5 AEC “Following each experiment, the column was regenerated and washed with a 1M NaOH solution until both the conductivity and UV absorbance at 280 nm stabilized, and then washed with PBS solution until both the conductivity and UV absorbance at 280 nm stabilized. Bubbles in the chromatographic column were checked for, and once it was confirmed that there were no issues, the next set of experiments commenced to maintain independence between experimental groups. This procedure was replicated three times for each type of gel filtration medium.”(Lines 152-158)
We are also surprised by the appearance of the current chromatograms. We speculate that it may be due to the rigidity instability of the chromatographic medium, or the uneven particle size and pore size. However, it is worth noting that these products have all passed quality control. What's even more surprising is that when the liquids collected according to the current process were subjected to subsequent testing, there was no statistically significant difference in antigen content.
To investigate this phenomenon, detailed analysis of the material could be considered in subsequent experiments, such as using a Malvern laser diffraction instrument to measure its particle size distribution, using the pressure-flow rate method to test its pressure resistance, and using Fourier transform infrared spectroscopy to detect its chemical bonds or functional group information. We would also like to ask for your thoughts on this phenomenon.
We look forward to your further suggestions.
Comments 19:Line 296-298, "Q Singarose demonstrates the best performance, exhibiting well-defined spectra with consistent peak times and no impurities", provide evidence of this.
Response 19:During the early stages of process development and formal production, we utilized GE's Sepharose 6FF and Capto-Q as the chromatography media. By comparing the chromatograms from previous experiments, we found that the peak shape of Q Singarose closely resembled that of Capto-Q, leading us to draw this conclusion.
Unfortunately, due to clinical application requirements, this portion of the experimental results must remain confidential, and we are unable to make a formal publication. We sincerely appreciate your interest in and support of our research.
Comments 20:Line 301-302, "in the final pure solution obtained through AEC purification", you need to be specific, "a sample collected from 400 to 450 mL of elution volume", something like that.
Response 20:I'm very sorry, the description in the paper is not clear and specific enough. We will annotate the corresponding figures to make the paper more clear and understandable.
Figure 5. Purification of the initial purification liquids obtained through SEC purification by AEC chromatography. Different media were used respectively: (A) Maxtar Q; (B) Q Singarose; (C) Diamond Q; (D) Q Focurose. The column specifications were XK 50/30, with 0.5 M NaCl as the elution buffer at a flow rate of 5 mL/min. Two purple vertical lines indicate the range where sample collection can and must begin (0.72-0.82 CV), while two yellow vertical lines indicate the range where sample collection can and must end (0.84-0.97 CV).(Line 363)
Thank you for your valuable suggestions.
Comments 21:Figures 6 and 7, nothing can be read in these plots. Response 21:Thank you for your valuable feedback. We acknowledge that the readability of these plots needs improvement.
Figure 6. The impact of different batches of hydrolytic concentrated solution on the quality parameters. Show the effects of the batch on the antigen recovery rate, specific activity, protein content, antigen content, HCP content, Vero DNA content, and BSA content in the final purified liquid obtained after AEC. Each point represents the results of the final purified liquid detected in a set of experiments, and the unbiased estimate of the overall mean (mean)is also indicated on the graph.(Line 391)
Figure 7. The impact of different AEC chromatograpy medium on the quality parameters. Show the effects of different AEC chromatography medias on the antigen recovery rate, specific activity, protein content, antigen content, HCP content, Vero DNA content, and BSA content in the final purified liquid obtained after AEC. Each point represents the results of the final purified liquid detected in a set of experiments, and the unbiased estimate of the overall mean (mean)is also indicated on the graph.(Line 419)
Thank you for bringing this to our attention.
Comments 22:A lot of analyses are provided. Despite this, regular yield and purity must be calculated. What is the antigen yield? What are the initial and final purities of the antigen? The resulting antigen concentration is enough for immunization?
Response 22:Thank you very much for your attention and questions. In our research, we have indeed conducted a series of analyses and calculated the regular yield and purity in the early stages of process development. However, our research primarily focuses on the robustness of the evaluated and developed processes, so these parameters were not taken into consideration when designing the experiments.
In our previous purity tests, we found that the correlation between the antigenicity and immunogenicity of the inactivated COVID-19 vaccine and its purity was not significant. Even when there were non-single peaks observed in the purity testing using HPLC, we still observed good protective effects. This may be due to the protective efficacy of the S protein fragments and RBD fragments present in the incomplete particles of the SARS-CoV-2 virus.
Furthermore, we have also tested the residual amounts of HCP (host cell proteins) and DNA. They comply with safety requirements as regulated by law. Purity testing was not conducted in subsequent clinical submission materials.
The final antigen concentration is sufficient for immunization, and we can dilute it as needed and add components such as adjuvants and preservatives to produce the final vaccine product.
At the same time, I understand your interest in yield. Here are the yield data observed in our experiments (insert chart).
If you have more questions or need further information on other aspects, please feel free to let me know. Thank you very much for your support and understanding.
Comments 23:A lot of analyses are provided. Despite this, regular yield and purity must be calculated. What is the antigen yield? What are the initial and final purities of the antigen? The resulting antigen concentration is enough for immunization?
Response 23:Thank you very much for your attention and questions.
1.Purity: In our research, we indeed did not test for purity, as our focus was primarily on the robustness of the evaluated and developed processes. Therefore, purity was not taken into consideration when designing the experiments. In the earlier stages of process development, we found that the correlation between the antigenicity, immunogenicity, and purity of the inactivated COVID-19 vaccine was not significant. Even when non-single peaks were observed in the purity testing using HPLC, we still observed good protective effects. This may be due to the protective efficacy of the S protein fragments and RBD fragments present in the incomplete particles of the SARS-CoV-2 virus. Additionally, we have tested the residual amounts of HCP (host cell proteins) and DNA. They comply with safety requirements as regulated by law. Purity testing was not conducted in subsequent clinical submission materials.
2.Concentration: The final antigen concentration is sufficient for immunization, and we can dilute it as needed and add components such as adjuvants and preservatives to produce the final vaccine product.
3.Yield: Yield can be calculated through antigen recovery rates.
If you have more questions or need further information on other aspects, please feel free to let me know. Thank you very much for your support and understanding. Comments 24:For AEX, it is not clear what peak did you compare between columns. A difficult task considering the chromatograms of Maxtar and Diamond.
Response 24:Thank you for your insightful comments.We have made revisions to the images and clarified the methodology.
Figure 5. Purification of the initial purification liquids obtained through SEC purification by AEC chromatography. Different media were used respectively: (A) Maxtar Q; (B) Q Singarose; (C) Diamond Q; (D) Q Focurose. The column specifications were XK 50/30, with 0.5 M NaCl as the elution buffer at a flow rate of 5 mL/min. Two purple vertical lines indicate the range where sample collection can and must begin (0.72-0.82 CV), while two yellow vertical lines indicate the range where sample collection can and must end (0.84-0.97 CV).(Line 363)
“The specific method for collecting samples is as follows, in accordance with the current practices in the workshop. The collection of samples can begin at 0.72 CV, which means no substances should be collected before this point, even if there is material detected at UV 280 nm. The collection must start at 0.82 CV, even if no material is detected at UV 280 nm at this time. The collection of samples can be completed at 0.84 CV, which means it must continue until this point, even if no material flows out at UV 280 nm before this. The collection of samples must be completed at 0.97 CV, even if there is still material flowing out at UV 280 nm at this time.”(Line 176-183)
|
||||||||||||||||||||||||||||||||||||||||||||||||||||||||||||||||||||||||
4. Response to Comments on the Quality of English Language |
||||||||||||||||||||||||||||||||||||||||||||||||||||||||||||||||||||||||
Point 1:English language fine. No issues detected |
||||||||||||||||||||||||||||||||||||||||||||||||||||||||||||||||||||||||
Response 1:Thank you for your feedback regarding the quality of the English language. I appreciate your comment and I'm glad to hear that there were no issues detected. If you have any further questions or concerns, please feel free to let me know.
|
||||||||||||||||||||||||||||||||||||||||||||||||||||||||||||||||||||||||
5. Additional clarifications |
||||||||||||||||||||||||||||||||||||||||||||||||||||||||||||||||||||||||
I have reviewed the comments and suggestions provided by the journal editor and reviewer, and I would like to express my gratitude for their thorough review of my work. After careful consideration, I do not have any additional clarifications to provide at this time. I believe that the manuscript adequately addresses the research questions and presents the findings in a clear and concise manner. However, I am open to further discussion or clarification if requested by the editor or reviewer. |

Reviewer 3 Report
Comments and Suggestions for Authors
The robustness and efficiency of the size exclusion chromatography (SEC) and anion exchange chromatography (AEC) for purification of the inactivated COVID-19 vaccine was studied. Four gel filtration chromatography media and four ion exchange chromatography media was utilized to purify the hydrolytic concentrated solution from three different batches. Various quality parameters was examined and statistical analysis was conducted to assess the purification efficiencies. It has been shown that the process exhibits satisfactory stability during both the gel filtration and ion exchange chromatography stages. It was not found statistically significant differences in purification efficiency among the four gel filtration chromatography media used. On the other hand, statistically significant differences in purification efficiency among the four ion exchange chromatography media was observed. It emphasizes the significance of media selection in optimizing purification efficiency.
The notes
1) The beginning of the article is too pathetic
2) The font in the pictures is too small
3) Table 3 shows Maxtar Q, Q Singarose, Diamond Q and Q Focurose as media for AEC chromatography columns, but in the figure 5 Chromstar 6FF is indicated instead of Maxtar Q.
4) There are two almost identical phrases in the text:
….. Consequently, in this experiment, only the first major peak corresponding to the virus was collected for subsequent analytical testing. (Line 198)
….. Consequently, in this specific experiment, only the first elution peak was collected for subsequent analysis. (Line 286)
The first case refers to gel filtration chromatography and indeed, in Figure 1 three peaks are clearly visible.
The second case refers to anion exchange chromatography, but the first elution peak is not visible in the corresponding figures
Author Response
Please see the attachment.
For research article
Response to Reviewer 3 Comments
|
||
1. Summary |
|
|
Thank you for your time and effort in reviewing this manuscript. Below, you will find our detailed responses to your comments, along with the corresponding revisions and corrections clearly highlighted or tracked in the resubmitted files. We greatly appreciate your thorough evaluation and valuable feedback.
|
||
2. Questions for General Evaluation |
Reviewer’s Evaluation |
Response and Revisions |
Does the introduction provide sufficient background and include all relevant references? |
Yes |
|
Are all the cited references relevant to the research? |
Yes |
|
Is the research design appropriate? |
Yes |
|
Are the methods adequately described? |
|
We have provided our corresponding response in the point-by-point response letter. |
Are the results clearly presented? |
Can be improved |
We have provided our corresponding response in the point-by-point response letter. |
Are the conclusions supported by the results? |
Yes |
|
3. Point-by-point response to Comments and Suggestions for Authors |
||
Comments 1: 1) The beginning of the article is too pathetic
|
||
Response 1: Thank you for your feedback. We appreciate your comment and would like to assure you that we have made revisions to the introduction of the article. “Abstract: Background: The vaccine production at a large scale requires the downstream processing focused on robustness, efficiency and cost-effectiveness. Methods: Three batches of COVID-19 Omicron strain hydrolytic concentrated solutions were chosen to assess the current vaccine production process's robustness. We used four gel filtration chromatography media (Chromstar 6FF, Singarose FF, Bestarose 6B, and Focurose 6FF) and four ion exchange chromatography media (Maxtar Q, Q Singarose, Diamond Q, and Q Focurose) to evaluate their impact on vaccine purification. We assessed vaccine quality by analyzing total protein content, antigen content, residual Vero cell DNA, residual Vero cell protein (HCP), and residual bovine serum albumin (BSA). We also calculated antigen recovery rate and specific activity. Statistical analysis was conducted to evaluate the process robustness and the chromatography media's purification effects. Results: The statistical analysis found no significant differences in antigen recovery (p = 0.10), Vero HCP residue (p = 0.59), Vero DNA residue (p = 0.28), and BSA residue (p = 0.97) among the three batches of hydrolytic concentrated solutions processed as per the current method. Nevertheless, a significant difference (p < 0.001) was observed in antigen content. Conclusions: Our study demonstrated the remarkable robustness of the current downstream process for producing WIBP-CorV vaccines, which is capable of adapting to different batches of hydrolytic concentrated solutions and various chromatography media. This research is critical for inactivated SARS-CoV-2 vaccine production and offers a potential template for purifying other viruses.”(Lines 10-28)
“Abstract: Background: Large-scale vaccine production requires downstream processing that focuses on robustness, efficiency, and cost-effectiveness.Methods: To assess the robustness of the current vaccine production process, three batches of COVID-19 Omicron BA.1 strain hydrolytic concentrated solutions were selected. Four gel filtration chromatography media (Chromstar 6FF, Singarose FF, Bestarose 6B, and Focurose 6FF) and four ion exchange chromatography media (Maxtar Q, Q Singarose, Diamond Q, and Q Focurose) were used to evaluate their impact on vaccine purification. The quality of the vaccine was assessed by analyzing total protein content, antigen content, residual Vero cell DNA, residual Vero cell protein, and residual bovine serum albumin (BSA). Antigen recovery rate and specific activity were also calculated. Statistical analysis was conducted to evaluate process robustness and the purification effects of the chromatography media.Results: The statistical analysis revealed no significant differences in antigen recovery (p = 0.10), Vero HCP residue (p = 0.59), Vero DNA residue (p = 0.28), and BSA residue (p = 0.97) among the three batches of hydrolytic concentrated solutions processed according to the current method. However, a significant difference (p < 0.001) was observed in antigen content.Conclusions: The study demonstrated the remarkable robustness of the current downstream process for producing WIBP-CorV vaccines. This process can adapt to different batches of hydrolytic concentrated solutions and various chromatography media. The research is crucial for the production of inactivated SARS-CoV-2 vaccines and provides a potential template for purifying other viruses.”(Lines 10-27)
Thank you for your valuable input, and we hope the revisions will meet your expectations.
|
||
Comments 2:2) The font in the pictures is too small Response 2: Thank you for your feedback. We appreciate your concern regarding the font size in the pictures. We have already made the necessary adjustments to ensure that the images are clearly legible and meet the required standards.
Figure 1. Purification of three batches of the hydrolytic concentrated solution by SEC chromatography. Different media were used respectively: (A) Chromstar 6FF; (B) Singarose 6FF; (C) Bestarose 6B; (D) Focurose 6FF. The column specifications were XK 16/100, with PBS as the mobile phase at a flow rate of 1 mL/min.(Line 256)
Figure 2. The peak areas of the virus peaks in the UV280 nm chromatograms.Statistics were conducted according to different grouping methods: (A) according to the batch of hydrolytic concentrated solution used; (B) according to the chromatograpy medium used.(Line 275)
Figure 3. The impact of different batches of hydrolytic concentrated solution on the quality parameters. Show the effects of the batch on the antigen recovery rate, specific activity, protein content, antigen content, HCP content, Vero DNA content, and BSA content in the initial purified liquid obtained after SEC. Each point represents the results of the initial pure liquid detected in a set of experiments, and the unbiased estimate of the overall mean (mean)is also indicated on the graph.(Line 309)
Figure 4. The impact of different chromatography medias on the quality parameters. Show the effects of four different chromatography medias on the antigen recovery rate, specific activity, protein content, antigen content, HCP content, Vero DNA content, and BSA content in the initial purified liquid obtained after SEC.Each point represents the results of the initial pure liquid detected in a set of experiments, and the unbiased estimate of the overall mean (mean)is also indicated on the graph.(Line 342)
Figure 5. Purification of the initial purification liquids obtained through SEC purification by AEC chromatography. Different media were used respectively: (A) Maxtar Q; (B) Q Singarose; (C) Diamond Q; (D) Q Focurose. The column specifications were XK 50/30, with 0.5 M NaCl as the elution buffer at a flow rate of 5 mL/min. Two purple vertical lines indicate the range where sample collection can and must begin (0.72-0.82 CV), while two yellow vertical lines indicate the range where sample collection can and must end (0.84-0.97 CV).(Line 363)
Figure 6. The impact of different batches of hydrolytic concentrated solution on the quality parameters. Show the effects of the batch on the antigen recovery rate, specific activity, protein content, antigen content, HCP content, Vero DNA content, and BSA content in the final purified liquid obtained after AEC. Each point represents the results of the final purified liquid detected in a set of experiments, and the unbiased estimate of the overall mean (mean)is also indicated on the graph.(Line 391)
Figure 7. The impact of different AEC chromatograpy medium on the quality parameters. Show the effects of different AEC chromatography medias on the antigen recovery rate, specific activity, protein content, antigen content, HCP content, Vero DNA content, and BSA content in the final purified liquid obtained after AEC. Each point represents the results of the final purified liquid detected in a set of experiments, and the unbiased estimate of the overall mean (mean)is also indicated on the graph.(Line 419)
Figure 8. Correlation Analysis Matrix Figure. The left panel shows the correlation analysis between the quality parameters of the samples after SEC purification, the SEC column-related parameters, and cell culture data. The right panel displays the correlation analysis between the quality parameters of the samples after AEC purification, the AEC column-related parameters, and cell culture data. The abbreviations on the diagonal are as follows: Pro: Protein; Anti: Antigen; HCP: Vero Host Cell Protein; DNA: Vero DNA; BSA: Bovine Albumin; RR: Antigen Recovery Rate; CD: Cell Density; HETP: Height Equivalent of Theoretical Plate; Asym: Asymmetry; PA: Peak Area. In the figure, red represents negative correlation, blue represents positive correlation, and the darker the color, the greater the correlation coefficient.(Line 440)
|
||
Comments 3:Table 3 shows Maxtar Q, Q Singarose, Diamond Q and Q Focurose as media for AEC chromatography columns, but in the figure 5 Chromstar 6FF is indicated instead of Maxtar Q.
|
||
4. Response to Comments on the Quality of English Language |
||
Point 1:I am not qualified to assess the quality of English in this paper. Response 1:Thank you for your feedback regarding the quality of the English language. If you have any further questions or concerns, please feel free to let me know.
|
||
5. Additional clarifications |
||
I have reviewed the comments and suggestions provided by the journal editor and reviewer, and I would like to express my gratitude for their thorough review of my work. After careful consideration, I do not have any additional clarifications to provide at this time. I believe that the manuscript adequately addresses the research questions and presents the findings in a clear and concise manner. However, I am open to further discussion or clarification if requested by the editor or reviewer. |

Reviewer 4 Report
Comments and Suggestions for Authors
This is relatively solid work, optimizing the various steps of purification of SARS-CoV-2 antigen for use as vaccines. The steps are conducted well, presented comprehensively for the readers. Importantly, the authors, utilizing three distinct batches of the COVID-19 strains, considering the variations in the properties and quantities of the hydrolytic concentrated solution among different batches, which covered a significant element of diversity. The purification process was assessed by measuring multiple key quality parameters. Lastly, the statistical analyses are thorough.
Nevertheless, I will mention several mostly technical points for the authors to address:
1) The authors have used chromatography media from four different companies, but they are Chinese companies in China. This may make it difficult for scientists in other nations to follow these procedures.
2) All figures are impossible to read, due to these extremely small size.
3) There is absolutely no mention about the CoVID-19 RNA vaccine that was immensely successful. In the Introduction, the authors should state their rational for paying attention to whole virus vaccine.
4) MDPI is an international publishing agency, and its journals are expected to be read by all nations, not just China. The good manufacturing practices in all countries require the purity screenings mentioned here, not only by the “Chinese Pharmacopoeia” (Lines 55/56, 354). I suggest modifying these sentences to a universal one, deleting references to the Pharmacopoeia.
For example, now there are two sentences: “While SEC excels in removing impurities such as proteins, it can be challenged by the efficient removal of host cell DNA (hcDNA), often due to adsorption interactions between the virus and hcDNA [17]. The Chinese Pharmacopoeia imposes strict regulations on the level of hcDNA residue, prompting the use of anion exchange chromatography (AEC) for effective hcDNA clearance [18].”
They can be combined into one: “While SEC excels in removing impurities such as proteins, it can be challenged by the need for efficient removal of host cell DNA, thus prompting the use of anion exchange chromatography (AEC) for effective hcDNA clearance [17,18].”
5) Line 50: “fillers” may be “filters”.
Comments on the Quality of English LanguageRead it carefully. Some places can be improved. This is a minor point.
Author Response
Please see the attachment.
For research article
Response to Reviewer 4 Comments
|
||
1. Summary |
|
|
Thank you for your time and effort in reviewing this manuscript. Below, you will find our detailed responses to your comments, along with the corresponding revisions and corrections clearly highlighted or tracked in the resubmitted files. We greatly appreciate your thorough evaluation and valuable feedback.
|
||
2. Questions for General Evaluation |
Reviewer’s Evaluation |
Response and Revisions |
Does the introduction provide sufficient background and include all relevant references? |
Can be improved |
We have provided our corresponding response in the point-by-point response letter. |
Are all the cited references relevant to the research? |
Yes |
|
Is the research design appropriate? |
Yes |
|
Are the methods adequately described? |
Yes |
|
Are the results clearly presented? |
Yes |
|
Are the conclusions supported by the results? |
Yes |
|
3. Point-by-point response to Comments and Suggestions for Authors |
||
Comments 1: 1) The authors have used chromatography media from four different companies, but they are Chinese companies in China. This may make it difficult for scientists in other nations to follow these procedures.
|
||
Response 1: Thank you for reviewing our paper. Regarding the issue you raised about the chromatography column packing materials, we would like to emphasize the following points:
1.International production standards: The production standards at our research institution are aligned with international practices, ensuring the reproducibility and universality of our research.
2.International benchmarking of technical parameters: Although the manufacturers of the chromatography packing materials we used are based in China, their technical parameters are benchmarked against internationally recognized products such as Sepharose 6FF and Capto Q.
3.We are acutely aware of the importance of the universality and reproducibility of scientific research in advancing global scientific progress. Incorporating commonly used chromatography packing materials from international sources can enhance the international applicability of our research. During the earlier stage of process development, we exclusively utilized Sepharose 6FF and Capto Q; however, due to clinical application requirements, this data cannot be publicly disclosed.
Once again, we appreciate your valuable feedback, and we will carefully consider and implement the necessary improvements.
|
||
Comments 2:2) All figures are impossible to read, due to these extremely small size.
|
||
Response 2: Thank you for your feedback. We appreciate your concern regarding the font size in the pictures. We have already made the necessary adjustments to ensure that the images are clearly legible and meet the required standards.
Figure 1. Purification of three batches of the hydrolytic concentrated solution by SEC chromatography. Different media were used respectively: (A) Chromstar 6FF; (B) Singarose 6FF; (C) Bestarose 6B; (D) Focurose 6FF. The column specifications were XK 16/100, with PBS as the mobile phase at a flow rate of 1 mL/min.(Line 256)
Figure 2. The peak areas of the virus peaks in the UV280 nm chromatograms.Statistics were conducted according to different grouping methods: (A) according to the batch of hydrolytic concentrated solution used; (B) according to the chromatograpy medium used.(Line 275)
Figure 3. The impact of different batches of hydrolytic concentrated solution on the quality parameters. Show the effects of the batch on the antigen recovery rate, specific activity, protein content, antigen content, HCP content, Vero DNA content, and BSA content in the initial purified liquid obtained after SEC. Each point represents the results of the initial pure liquid detected in a set of experiments, and the unbiased estimate of the overall mean (mean)is also indicated on the graph.(Line 309)
Figure 4. The impact of different chromatography medias on the quality parameters. Show the effects of four different chromatography medias on the antigen recovery rate, specific activity, protein content, antigen content, HCP content, Vero DNA content, and BSA content in the initial purified liquid obtained after SEC.Each point represents the results of the initial pure liquid detected in a set of experiments, and the unbiased estimate of the overall mean (mean)is also indicated on the graph.(Line 342)
Figure 5. Purification of the initial purification liquids obtained through SEC purification by AEC chromatography. Different media were used respectively: (A) Maxtar Q; (B) Q Singarose; (C) Diamond Q; (D) Q Focurose. The column specifications were XK 50/30, with 0.5 M NaCl as the elution buffer at a flow rate of 5 mL/min. Two purple vertical lines indicate the range where sample collection can and must begin (0.72-0.82 CV), while two yellow vertical lines indicate the range where sample collection can and must end (0.84-0.97 CV).(Line 363)
Figure 6. The impact of different batches of hydrolytic concentrated solution on the quality parameters. Show the effects of the batch on the antigen recovery rate, specific activity, protein content, antigen content, HCP content, Vero DNA content, and BSA content in the final purified liquid obtained after AEC. Each point represents the results of the final purified liquid detected in a set of experiments, and the unbiased estimate of the overall mean (mean)is also indicated on the graph.(Line 391)
Figure 7. The impact of different AEC chromatograpy medium on the quality parameters. Show the effects of different AEC chromatography medias on the antigen recovery rate, specific activity, protein content, antigen content, HCP content, Vero DNA content, and BSA content in the final purified liquid obtained after AEC. Each point represents the results of the final purified liquid detected in a set of experiments, and the unbiased estimate of the overall mean (mean)is also indicated on the graph.(Line 419)
Figure 8. Correlation Analysis Matrix Figure. The left panel shows the correlation analysis between the quality parameters of the samples after SEC purification, the SEC column-related parameters, and cell culture data. The right panel displays the correlation analysis between the quality parameters of the samples after AEC purification, the AEC column-related parameters, and cell culture data. The abbreviations on the diagonal are as follows: Pro: Protein; Anti: Antigen; HCP: Vero Host Cell Protein; DNA: Vero DNA; BSA: Bovine Albumin; RR: Antigen Recovery Rate; CD: Cell Density; HETP: Height Equivalent of Theoretical Plate; Asym: Asymmetry; PA: Peak Area. In the figure, red represents negative correlation, blue represents positive correlation, and the darker the color, the greater the correlation coefficient.(Line 440)
Comments 3:There is absolutely no mention about the CoVID-19 RNA vaccine that was immensely successful. In the Introduction, the authors should state their rational for paying attention to whole virus vaccine.
Response 3: We deeply regret the oversight of not including the highly successful CoVID-19 RNA vaccine in our manuscript. We recognize the importance of addressing this in the Introduction section. While our focus has been on whole virus vaccines, we have closely monitored advancements in various other platforms, such as mRNA, viral vectors, and recombinant proteins, during the development of our inactivated vaccine. Furthermore, researchers at our institution are actively involved in developing mRNA vaccines for both the novel coronavirus and influenza virus.
The reason for choosing an inactivated vaccine is twofold:
1.The technology for inactivated vaccines is highly mature, and our laboratory has extensive experience in this area. During the initial stages of the pandemic, mRNA vaccine technology was not widely available. Subsequently, we have also conducted research on mRNA vaccines in our ongoing studies.
2.The safety of inactivated vaccines is extremely reliable due to viral inactivation, resulting in minimal side effects.
We will ensure that the revised manuscript includes relevant content addressing these points. We value your input and thank you for bringing this matter to our attention.
“The reason for choosing the inactivated vaccine technology was that during its development, it was still in the early stages of the pandemic when mRNA vaccine technology had not yet been widely promoted. Therefore, we opted for this technology at that time. Additionally, we closely monitored various new vaccine technologies. Subsequently, we also pursued the development of mRNA vaccines for both influenza and the novel coronavirus.”(Line 66-71)
Comments 4:MDPI is an international publishing agency, and its journals are expected to be read by all nations, not just China. The good manufacturing practices in all countries require the purity screenings mentioned here, not only by the “Chinese Pharmacopoeia” (Lines 55/56, 354). I suggest modifying these sentences to a universal one, deleting references to the Pharmacopoeia.
For example, now there are two sentences: “While SEC excels in removing impurities such as proteins, it can be challenged by the efficient removal of host cell DNA (hcDNA), often due to adsorption interactions between the virus and hcDNA [17]. The Chinese Pharmacopoeia imposes strict regulations on the level of hcDNA residue, prompting the use of anion exchange chromatography (AEC) for effective hcDNA clearance [18].”
They can be combined into one: “While SEC excels in removing impurities such as proteins, it can be challenged by the need for efficient removal of host cell DNA, thus prompting the use of anion exchange chromatography (AEC) for effective hcDNA clearance [17,18].”
Response 4:Thank you for your valuable feedback. We appreciate your suggestion about modifying the sentences to make them more universal and applicable to all countries. We definitely take your suggestion into consideration and modify the manuscript accordingly. The revised sentence will be as follows:
“While SEC excels in removing impurities such as proteins, it can be challenged by the efficient removal of host cell DNA (hcDNA), often due to adsorption interactions between the virus and hcDNA [17]. The Chinese Pharmacopoeia imposes strict regulations on the level of hcDNA residue, prompting the use of anion exchange chromatography (AEC) for effective hcDNA clearance [18]. “
“While SEC excels in removing impurities such as proteins, it can be challenged by the need for efficient removal of host cell DNA, thus prompting the use of anion exchange chromatography (AEC) for effective hcDNA clearance [17,18]. “(Lines 54-57)
We modify the manuscript to make it more international in scope.
We are thankful for your insightful input and look forward to implementing the necessary changes.
Comments 5:Line 50: “fillers” may be “filters”.
Response 5:Thank you for your valuable feedback. We have carefully reviewed your suggestion and have made the necessary changes.
“Size exclusion chromatography (SEC), a technique based on the size of molecules, is particularly effective. It enables the separation of viruses of different sizes by selecting fillers with varying pore sizes [12]”(Line 50)
“Size exclusion chromatography (SEC), a technique based on the size of molecules, is particularly effective. It enables the separation of viruses of different sizes by selecting filters with varying pore sizes [12]”(Line 50-52)
We appreciate your attention to detail and thank you for helping us improve the quality of our work.
|
||
4. Response to Comments on the Quality of English Language |
||
Point 1:Minor editing of English language required |
||
Response 1: Thank you for your feedback. We will ensure that minor editing of the English language is carried out as per your suggestion. |
||
5. Additional clarifications |
||
I have reviewed the comments and suggestions provided by the journal editor and reviewer, and I would like to express my gratitude for their thorough review of my work. After careful consideration, I do not have any additional clarifications to provide at this time. I believe that the manuscript adequately addresses the research questions and presents the findings in a clear and concise manner. However, I am open to further discussion or clarification if requested by the editor or reviewer. |

Reviewer 5 Report
Comments and Suggestions for Authors
This study was designed to evaluate the robustness and efficiency of the current downstream process used to purify the inactivated COVID-19 vaccine. The model vaccine used was the Omicron BA.1 strain of SARS-CoV-2. Several key quality parameters were measured in both the initial and final purification samples to evaluate the effectiveness of the purification process. As a result, the authors confirm the stability and reliability of the current downstream COVID-19 inactivated vaccine purification process. The work has been thorough and in compliance with all the necessary validation criteria. At the same time, it remains unclear what new insights this study adds to the existing body of scientific knowledge.
Other concerns. Screenshots of the instruments used to purify the vaccine are shown in Figures 1 through 7. As such, the data contained in these images is not available to those reading this article. The figure caption should also include explanatory information in addition to the figure title. The Correlation Analysis (see Section 3.3 and Figure 8) requires a more detailed description to understand.
Author Response
Please see the attachment.
For research article
Response to Reviewer 5 Comments
|
||
1. Summary |
|
|
Thank you for your time and effort in reviewing this manuscript. Below, you will find our detailed responses to your comments, along with the corresponding revisions and corrections clearly highlighted or tracked in the resubmitted files. We greatly appreciate your thorough evaluation and valuable feedback.
|
||
2. Questions for General Evaluation |
Reviewer’s Evaluation |
Response and Revisions |
Does the introduction provide sufficient background and include all relevant references? |
Yes |
|
Are all the cited references relevant to the research? |
Yes |
|
Is the research design appropriate? |
Yes |
|
Are the methods adequately described? |
Yes |
|
Are the results clearly presented? |
Must be improved |
We have provided our corresponding response in the point-by-point response letter. |
Are the conclusions supported by the results? |
Yes |
|
3. Point-by-point response to Comments and Suggestions for Authors |
||
Comments 1: This study was designed to evaluate the robustness and efficiency of the current downstream process used to purify the inactivated COVID-19 vaccine. The model vaccine used was the Omicron B.1.1.529 strain of SARS-CoV-2. Several key quality parameters were measured in both the initial and final purification samples to evaluate the effectiveness of the purification process. As a result, the authors confirm the stability and reliability of the current downstream COVID-19 inactivated vaccine purification process. The work has been thorough and in compliance with all the necessary validation criteria. At the same time, it remains unclear what new insights this study adds to the existing body of scientific knowledge.
|
||
Response 1:Thank you very much for your review of our paper and for your valuable feedback. We fully understand your concerns regarding the level of innovation in our study. We would like to clarify the innovative aspects of our research:
1.In-depth Detail: While our study adopts standard manufacturing processes, we conducted in-depth investigations using virus concentrates from different batches and chromatography resins from various manufacturers. This contributes to filling the gaps in detailed research within the relevant field.
2.Robustness Analysis: Our study primarily evaluates the robustness of the current purification process in producing inactivated SARS-CoV-2 B.1.1.529 strain strain vaccines. This is crucial in ensuring the reliability of the vaccine production process, as an ideal production process should maintain consistency across batches when dealing with different raw materials and equipment.
3.Practical Application Significance: Our work addresses challenges that may arise in real-world production environments. Other researchers can utilize the results of our experiment to select appropriate resins or attempt to expand the application of this production process to purify other viruses. This holds practical significance for optimizing vaccine production processes and problem-solving.
In summary, we believe that while our study utilizes standard process flows, the in-depth exploration of details and robustness analysis contribute to its innovation and practical applicability.
Once again, we appreciate your review and suggestions.
|
||
Comments 2: Other concerns. Screenshots of the instruments used to purify the vaccine are shown in Figures 1 through 7. As such, the data contained in these images is not available to those reading this article. The figure caption should also include explanatory information in addition to the figure title. The Correlation Analysis (see Section 3.3 and Figure 8) requires a more detailed description to understand.
|
||
Response 2: 1.Thank you for your feedback. We appreciate your concern regarding the font size in the pictures. We have already made the necessary adjustments to ensure that the images are clearly legible and meet the required standards.
Figure 1. Purification of three batches of the hydrolytic concentrated solution by SEC chromatography. Different media were used respectively: (A) Chromstar 6FF; (B) Singarose 6FF; (C) Bestarose 6B; (D) Focurose 6FF. The column specifications were XK 16/100, with PBS as the mobile phase at a flow rate of 1 mL/min.(Line 256)
Figure 2. The peak areas of the virus peaks in the UV280 nm chromatograms.Statistics were conducted according to different grouping methods: (A) according to the batch of hydrolytic concentrated solution used; (B) according to the chromatograpy medium used.(Line 275)
Figure 3. The impact of different batches of hydrolytic concentrated solution on the quality parameters. Show the effects of the batch on the antigen recovery rate, specific activity, protein content, antigen content, HCP content, Vero DNA content, and BSA content in the initial purified liquid obtained after SEC. Each point represents the results of the initial pure liquid detected in a set of experiments, and the unbiased estimate of the overall mean (mean)is also indicated on the graph.(Line 309)
Figure 4. The impact of different chromatography medias on the quality parameters. Show the effects of four different chromatography medias on the antigen recovery rate, specific activity, protein content, antigen content, HCP content, Vero DNA content, and BSA content in the initial purified liquid obtained after SEC.Each point represents the results of the initial pure liquid detected in a set of experiments, and the unbiased estimate of the overall mean (mean)is also indicated on the graph.(Line 342)
Figure 5. Purification of the initial purification liquids obtained through SEC purification by AEC chromatography. Different media were used respectively: (A) Maxtar Q; (B) Q Singarose; (C) Diamond Q; (D) Q Focurose. The column specifications were XK 50/30, with 0.5 M NaCl as the elution buffer at a flow rate of 5 mL/min. Two purple vertical lines indicate the range where sample collection can and must begin (0.72-0.82 CV), while two yellow vertical lines indicate the range where sample collection can and must end (0.84-0.97 CV).(Line 363)
Figure 6. The impact of different batches of hydrolytic concentrated solution on the quality parameters. Show the effects of the batch on the antigen recovery rate, specific activity, protein content, antigen content, HCP content, Vero DNA content, and BSA content in the final purified liquid obtained after AEC. Each point represents the results of the final purified liquid detected in a set of experiments, and the unbiased estimate of the overall mean (mean)is also indicated on the graph.(Line 391)
Figure 7. The impact of different AEC chromatograpy medium on the quality parameters. Show the effects of different AEC chromatography medias on the antigen recovery rate, specific activity, protein content, antigen content, HCP content, Vero DNA content, and BSA content in the final purified liquid obtained after AEC. Each point represents the results of the final purified liquid detected in a set of experiments, and the unbiased estimate of the overall mean (mean)is also indicated on the graph.(Line 419)
Figure 8. Correlation Analysis Matrix Figure. The left panel shows the correlation analysis between the quality parameters of the samples after SEC purification, the SEC column-related parameters, and cell culture data. The right panel displays the correlation analysis between the quality parameters of the samples after AEC purification, the AEC column-related parameters, and cell culture data. The abbreviations on the diagonal are as follows: Pro: Protein; Anti: Antigen; HCP: Vero Host Cell Protein; DNA: Vero DNA; BSA: Bovine Albumin; RR: Antigen Recovery Rate; CD: Cell Density; HETP: Height Equivalent of Theoretical Plate; Asym: Asymmetry; PA: Peak Area. In the figure, red represents negative correlation, blue represents positive correlation, and the darker the color, the greater the correlation coefficient.(Line 440)
2.With regard to the Correlation Analysis, we will enhance the explanation of this analysis to provide a clearer understanding.
“The figure vividly illustrates the correlations between various process parameters and quality parameters, aiding in the identification of crucial process parameters associated with quality. In the SEC results, we noted specific correlations between different factors. Particularly, we observed that the cell density of the virus harvest displayed negative correlations with both antigen content (R² = -0.89) and the area of the virus peak in the UV 280 nm spectra (R² = -0.65). Additionally, column efficiency exhibited a positive correlation with antigen recovery rate (R² = 0.58), while asymmetry demonstrated a negative correlation with antigen recovery rate (R² = -0.56). However, in the AEC correlation analysis, no significant correlations with process conditions related to antigen content or antigen recovery rate were observed. “(Lines 429-438)
Your thorough review of our manuscript is greatly appreciated, and we welcome the opportunity to enhance its accessibility and clarity. Your input is invaluable to us, and we are dedicated to addressing these concerns in the revised manuscript.
We eagerly anticipate your continued guidance and any further suggestions you may have.
|
||
4. Response to Comments on the Quality of English Language |
||
Point 1:English language fine. No issues detected |
||
Response 1: Thank you for your feedback regarding the quality of the English language. I appreciate your comment and I'm glad to hear that there were no issues detected. If you have any further questions or concerns, please feel free to let me know.
|
||
5. Additional clarifications |
||
I have reviewed the comments and suggestions provided by the journal editor and reviewer, and I would like to express my gratitude for their thorough review of my work. After careful consideration, I do not have any additional clarifications to provide at this time. I believe that the manuscript adequately addresses the research questions and presents the findings in a clear and concise manner. However, I am open to further discussion or clarification if requested by the editor or reviewer. |

Round 2
Reviewer 1 Report
Comments and Suggestions for Authors
Although the paper has low novelty, it can still serve as a practical reference to researchers in the field. The revised paper is acceptable for publication.
Reviewer 2 Report
Comments and Suggestions for Authors
The authors have addressed all my concerns.
Reviewer 5 Report
Comments and Suggestions for Authors
After corrections made the article might be published.